# Universality laws for Gaussian mixtures in generalized linear models

**Yatin Dandi, Ludovic Stephan, Florent Krzakala**
Idephics, EPFL, Switzerland

**Bruno Loureiro**
DI, Ecole Normale Superieure, Paris, France

**Lenka Zdeborova**
SPOC, EPFL, Switzerland

## Abstract

A recent line of work in high-dimensional statistics working under the Gaussian mixture hypothesis has led to a number of results in the context of empirical risk minimization, Bayesian uncertainty quantification, separation of kernel methods and neural networks, ensembling and fluctuation of random features. We provide rigorous proofs for the applicability of these results to a general class of datasets $(\boldsymbol{x}_i, y_i)_{i=1,\ldots,n}$ containing independent samples from mixture distribution $\sum_{c \in \mathcal{C}} \rho_c P_c^{\boldsymbol{x}}$. Specifically, we consider the hypothesis class of generalized linear models $\hat{y} = F(\boldsymbol{\Theta}^\top \boldsymbol{x})$ and investigate the asymptotic joint statistics of a family of generalized linear estimators $(\boldsymbol{\Theta}^{(1)}, \ldots, \boldsymbol{\Theta}^{(M)})$, obtained either from (a) minimizing an empirical risk $\hat{R}_n^{(m)}(\boldsymbol{\Theta}^{(m)}; \boldsymbol{X}, \boldsymbol{y})$ or (b) sampling from the associated Gibbs measure $\exp(-\beta n \hat{R}_n^{(m)}(\boldsymbol{\Theta}^{(m)}; \boldsymbol{X}, \boldsymbol{y}))$. Our main contribution is to characterize under which conditions the asymptotic joint statistics of this family depend (on a weak sense) only on the means and covariances of the class conditional feature distribution $P_c^{\boldsymbol{x}}$. This allows us to prove the universality of different quantities of interest, including training and generalization errors, as well as the geometrical properties and correlations of the estimators.

A recurrent topic in high-dimensional statistics is the investigation of the typical properties of signal processing and machine learning methods on synthetic, *i.i.d.* Gaussian data, a scenario often known under the umbrella of *Gaussian design* [1, 2, 3, 4, 5]. A less restrictive assumption arises when considering that many machine learning tasks deal with data partitioned into a fixed number of classes. In these cases, the data distribution is naturally described by a *mixture model*, where each sample is generated *conditionally* on the class. In other words: data is generated by first sampling the class assignment and *then* generating the input conditioned on the class. Arguably the simplest example of such distributions is that of a *Gaussian mixture*, which shall be our focus in this work.

Gaussian mixtures are a popular model in high-dimensional statistics since, besides being an universal approximator, they often lead to mathematically tractable problems. Indeed, a recent line of work has analyzed the asymptotic performance of a large class of machine learning problems in the proportional high-dimensional limit under the Gaussian mixture data assumption, see e.g. [6, 7, 8, 9, 10, 11, 12]. The goal of the present work is to show that this assumption, and the conclusions derived therein, are far more general than previously anticipated.

A recent line of work [13, 14], initiated by the work of [15] for Kernel matrices, posits that for generalized linear estimation on non-linear feature maps satisfying certain regularity conditions and a "one-dimensional CLT", the data distribution can be replaced by equivalent Gaussian data without affecting the training and generalization errors. This was recently proven by [16] for empirical

risk minimization under the setup of strongly convex objectives, and extended to a larger class of objectives by [17].

However, there is strong empirical evidence that Gaussian universality holds in a more general sense [18]. First, existing results rely on the assumption of a target function depending on linear projections in the latent or feature space. This excludes the rich class of classification on mixture distributions, where the target function is given by the label. For such distributions, a more appropriate equivalent distribution is given by a mixture of Gaussians. Such a "Gaussian mixture equivalence" has been conjectured and used in existing works, such as [11, 12] and was found to closely approximately classification on real datasets.

Furthermore, equivalence with mixtures of Gaussians has been observed to hold not only for training, generalization errors but other quantities of the estimators such as overlaps, variance, etc. For instance, [19, 20] empirically observed that the equivalence holds even while considering the joint distribution of multiple uncertainty estimators or ensembles of multiple random feature mappings. This suggests the equivalence of the distributions of the minimizers themselves.

Our results fill these gaps and provide rigorous justification for the universality in all the aforementioned works. Namely, we show that the joint statistics of multiple generalized estimators obtained either from ERM or sampling on a mixture model asymptotically agrees (in a weak sense) with the statistics of estimators from the same class trained on a Gaussian mixture model with matching first and second order moments. Our **main contributions** are as follows:

• Through a generalization of recent developments in the Gaussian equivalence principle [14, 17, 16], we prove the universality of empirical risk minimization and sampling for a generic mixture distribution and an equivalent mixture of Gaussians. In particular, we show that a Gaussian mixture observed through a random feature map is also a Gaussian mixture in the high-dimensional limit, a fact used for instance (without rigorous justification) in [11, 12, 21, 22].
• A consequence of our results is that, with conditions on the matrix weights, data generated by conditional Generative Adversarial Networks (cGAN) behave as a Gaussian mixture when observed through the prism of generalized linear models (kernels, feature maps, etc...), as illustrated in Figs 1 and 2. This further generalizes the work of [23] that only considered the universality of Gram matrices for GAN generated data through the prism of random matrix theory.
• We construct a unified framework involving multiple sets of parameters arising from simultaneous minimization of different objectives as well as sampling from Gibbs distributions defined by the empirical risk. Through the design of suitable reductions and a convexity-based argument, we establish conditions for the asymptotic universality of arbitrary functions of the set of minimizers or samples from different Gibbs distributions (Theorem 4). For instance, it includes ensembling [20]), and the uncertainty quantification and Bayesian setting assumed (without proof) in [19, 24].
• Finally, we show that for multi-class classification, the conditions leading to universality hold for a large class of functions of the minimizers, such as overlaps and sparsity measures, leading to the equivalence between their distributions of themselves, and provide a theorem for their weak convergence (Theorem 5).
• As a technical contribution of independent interest, our proof of Theorem 5 demonstrates a principled approach for leveraging existing results on the exact asymptotics for simple data distributions (such as for Gaussian mixture models in [12]) to prove the weak convergence and universality of the joint-empirical measure of the estimators and parameters (means, covariances) of the data-distribution.

**Related works** — Universality is an important topic in applied mathematics, as it motivates the scope of tractable mathematical models. It has been extensively studied in the context of random matrix theory [25, 26], signal processing problems [1, 2, 3, 27, 28, 29, 30, 31] and kernel methods [15, 32, 33]. Closer to us is the recent stream of works that investigated the Gaussian universality of the asymptotic error of generalized linear models trained on non-linear features, starting from single-layer random feature maps [34, 35, 16, 36, 37] and later extended to single-layer NTK [17] and deep random features [38]. These results, together with numerical observations that Gaussian universality holds for more general classes of features, led to the formulation of different *Gaussian equivalence* conjectures [13, 14, 18]. Our results build on the line of works on the proofs of these conjectures, through the use of the one-dimensional CLT (Central Limit Theorem), stated formally in [14] who proved it for random features of Gaussian data. We generalize this principle to a Gaussian equivalence conditioned on the cluster assignment in a mixture-model with a corresponding conditional 1d-CLT (Assumption 10).

A complementary line of research has investigated cases in which the distribution of the features is multi-modal, suggesting a Gaussian mixture universality class instead [39, 23, 40]. A bridge between these two lines of work has been recently investigated with random labels and teachers in [21, 22]. Our results provide rigorous extensions of Gaussian universality to the setups of mixture models as well as uncertainty quantificatio and ensembling.

# 1 Setting and models

Consider a supervised learning problem where the training data $(\boldsymbol{x}_i, y_i) \in \mathbb{R}^p \times \mathcal{Y}$, $i \in [n] := \{1, \cdots, n\}$ is drawn *i.i.d.* from a mixture distribution:

$$\boldsymbol{x}_i \sim \sum_{c \in \mathcal{C}} \rho_c P_c^{\boldsymbol{x}}, \qquad\qquad \mathbb{P}(c_i = c) = \rho_c, \qquad\qquad (1)$$

with $c_i$ a categorical random variable denoting the cluster assignment for the $i_{th}$ example $\mathbf{x}_i$. Let $\boldsymbol{\mu}_c, \boldsymbol{\Sigma}_c$ denote the mean and covariance of $P_c^{\boldsymbol{x}}$, and $k = |\mathcal{C}|$. Further, assume that the labels $y_i$ are generated from the following target function:

$$y_i(\boldsymbol{X}) = \eta(\boldsymbol{\Theta}_\star^\top \boldsymbol{x}_i, \varepsilon_i, c_i), \qquad\qquad (2)$$

where $\eta : \mathbb{R}^3 \to \mathbb{R}$ is a general label generating function, $\boldsymbol{\Theta}_\star \in \mathbb{R}^{k \times p}$ and $\varepsilon_i$ is an i.i.d source of randomness. It is important to stress that the class labels (2) are themselves not constrained to arise from a simple function of the inputs $\boldsymbol{x}_i$. For instance, the functional form in (2) includes the case where the labels are exclusively given by a function of the mixture index $y_i = \eta(c_i)$. This will allow us to handle complex targets, such as data generated using conditional Generative Adversarial Networks (cGANs).

In this manuscript, we will be interested in hypothesis classes defined by parametric predictors of the form $y_{\boldsymbol{\Theta}}(\boldsymbol{x}) = F(\boldsymbol{\Theta}^\top \boldsymbol{x})$, where $\boldsymbol{\Theta} \in \mathbb{R}^{k \times p}$ are the parameters and $F : \mathbb{R}^k \to \mathcal{Y}$ a possibly non-linear function. For a given loss function $\ell : \mathbb{R}^k \times \mathcal{Y} \to \mathbb{R}_+$ and regularization term $r : \mathbb{R}^{k \times p} \to \mathbb{R}_+$, define the (regularized) empirical risk over the training data:

$$\widehat{\mathcal{R}}_n(\boldsymbol{\Theta}; \boldsymbol{X}, \boldsymbol{y}) := \frac{1}{n} \sum_{i=1}^n \ell(\boldsymbol{\Theta}^\top \boldsymbol{x}_i, y_i) + r(\boldsymbol{\Theta}), \qquad\qquad (3)$$

where we have defined the feature matrix $\boldsymbol{X} \in \mathbb{R}^{p \times n}$ by stacking the features $\boldsymbol{x}_i$ column-wise and the labels $y_i$ in a vector $\boldsymbol{y} \in \mathcal{Y}^n$. In what follows, we will be interested in the following two tasks:

(i) Minimization: in a minimization task, the statistician's goal is to find a good predictor by minimizing the empirical risk (3), possibly over a constraint set $\mathcal{S}_p$:

$$\widehat{\boldsymbol{\Theta}}_{\text{erm}}(\boldsymbol{X}, \boldsymbol{y}) \in \underset{\boldsymbol{\Theta} \in \mathcal{S}_p}{\arg\min} \, \widehat{\mathcal{R}}_n(\boldsymbol{\Theta}; \boldsymbol{X}, \boldsymbol{y}), \qquad\qquad (4)$$

This encompasses diverse settings such as generalized linear models with noise, two-layer networks with a constant number of neurons and fixed second layer, mixture classification, but also the random label setting (with $\eta(\boldsymbol{\Theta}_\star^\top \boldsymbol{x}_i, \varepsilon_i, c_i) = \varepsilon_i$). In the following, we denote $\widehat{\mathcal{R}}_n^\star(\boldsymbol{X}, \boldsymbol{y}) := \min_{\boldsymbol{\Theta}} \widehat{\mathcal{R}}_n(\boldsymbol{\Theta}; \boldsymbol{X}, \boldsymbol{y})$

(ii) Sampling: here, instead of minimizing the empirical risk (3), the statistician's goal is to sample from a Gibbs distribution that weights different hypothesis according to their empirical error:

$$\boldsymbol{\Theta}_{\text{Bayes}}(\boldsymbol{X}, \boldsymbol{y}) \sim P_{\text{Bayes}}(\boldsymbol{\Theta}) \propto \exp\left(-\beta n \widehat{\mathcal{R}}_n(\boldsymbol{\Theta}; \boldsymbol{X}, \boldsymbol{y})\right) \mathrm{d}\mu(\boldsymbol{\Theta}) \qquad\qquad (5)$$

where $\mu$ is reference prior measure and $\beta > 0$ is a parameter known as the *inverse temperature*. Note that minimization can be seen as a particular example of sampling when $\beta \to \infty$, since in this limit the above measure peaks on the global minima of (4).

**Applications of interest—** So far, the setting defined above is quite generic, and the motivation to study this problem might not appear evident to the reader. Therefore, we briefly discuss a few scenarios of interest which are covered by this model.

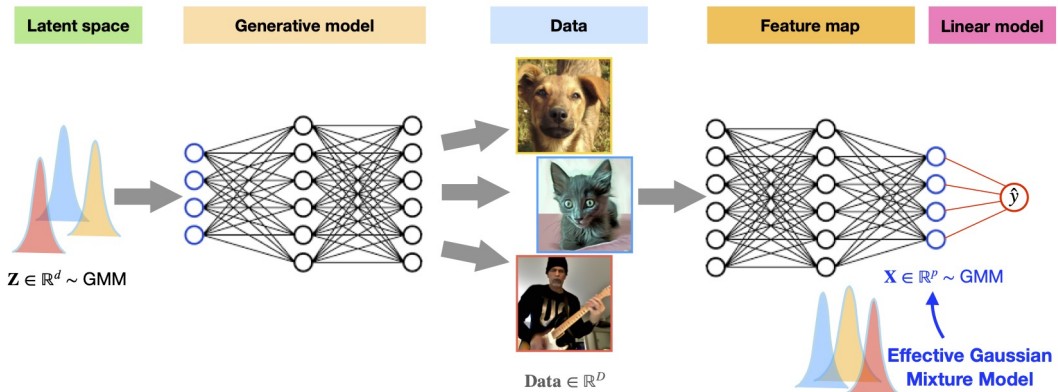

Figure 1: Illustration of Corollary 2: high-dimensional data generated by generative neural networks starting from a mixture of Gaussian in latent space ($\mathbf{z} \in \mathbb{R}^H$) are (with conditions on the weights matrices) equivalent, in high-dimension and for generalized linear models, to data sampled from a Gaussian mixture. A concrete example is shown in Fig. 2.

(i) *Conditional GANs (cGANs):* These were introduced by [41] as a generative model to learn mixture distributions. Once trained in samples from the target distribution, they define a function $\Psi$ that maps Gaussian mixtures (defining the latent space) to samples from the target mixture that preserve the label structure. In other words, conditioned on the label:

$$\forall c \in \mathcal{C}, \qquad \mathbf{z} \sim \mathcal{N}(\boldsymbol{\mu}_c, \boldsymbol{\Sigma}_c) \mapsto \mathbf{x}_c = \Psi(\mathbf{z}, c) \sim P_c^{\boldsymbol{x}} \tag{6}$$

The connection to model (1) is immediate. This scenario was extensively studied by [39, 23, 40], and is illustrated in Fig. 1. In Fig. 2 we report on a concrete experiment with a cGAN trained on the fashion-MNIST dataset.

(ii) *Multiple objectives:* Our framework also allows to characterize the joint statistics of estimators $(\boldsymbol{\Theta}_1, \ldots, \boldsymbol{\Theta}_M)$ obtained from empirical risk minimization and/or sampling from different objective functions $\hat{R}_n^m$ defined on the same training data $(\boldsymbol{X}, \boldsymbol{y})$. This can be of interest in different scenarios. For instance, [19, 24] has characterized the correlation in the calibration of different uncertainty measures of interest, e.g. last-layer scores and Bayesian training of last-layer weights. This crucially depends on the correlation matrix $\hat{\boldsymbol{\Theta}}_{\mathrm{erm}} \boldsymbol{\Theta}_{\mathrm{Bayes}}^{\top} \in \mathbb{R}^{k \times k}$ which fits our framework.

(iii) *Ensemble of features:* Another example covered by the multi-objective framework above is that of ensembling. Let $(\boldsymbol{z}_i, y_i) \in \mathbb{R}^d \times \mathcal{Y}$ denote some training data from a mixture model akin to (1). A popular ensembling scheme often employed in the context of deep learning [42] is to take a family of $M$ feature maps $\boldsymbol{z}_i \mapsto \boldsymbol{x}_i^{(m)} = \varphi_m(\boldsymbol{z}_i)$ (e.g. neural network features trained from different random initialization) and train $M$ independent learners:

$$\hat{\boldsymbol{\Theta}}_{\mathrm{erm}}^{(m)} \in \underset{\boldsymbol{\Theta} \in \mathcal{S}_p}{\arg\min} \frac{1}{n} \sum_{i=1}^{n} \ell(\boldsymbol{\Theta}^{\top} \boldsymbol{x}_i^{(m)}, y_i) + r(\boldsymbol{\Theta}) \tag{7}$$

Prediction on a new sample $\boldsymbol{z}$ is then made by ensembling the independent learners, e.g. by taking their average $\hat{\boldsymbol{y}} = 1/M \sum_{m=1}^{M} \hat{\boldsymbol{\Theta}}_{\mathrm{erm}}^{(m)\top} \varphi_m(\boldsymbol{z})$. A closely related model was studied in [43, 44, 20].

Note that in all the applications above, having the labels depending on the features $\boldsymbol{X}$ would not be natural, since they are either generated from a latent space, as in $(i)$, or chosen by the statistician, as in $(ii), (iii)$. Indeed, in these cases the most natural label model is given by the mixture index $y = c$ itself, which is a particular case of (2). This highlights the flexibility of our target model with respect to prior work [17]. Instead, [16] assumes that the target is a function of a *latent variable*, which would correspond to a mismatched setting. The discussion here can be generalized also to this case, but require an additional assumption discussed in Appendix B.

**Universality —** Given these tasks, the goal of the statistician is to characterize different statistical properties of these predictors. These can be, for instance, point performance metrics such as the empirical and population risks, or uncertainty metrics such as the calibration of the predictor or moments of the posterior distribution (5). These examples, as well as many different other quantities of interest, are functions of the joint statistics of the pre-activations $(\boldsymbol{\Theta}_{\star}^{\top} \boldsymbol{x}, \boldsymbol{\Theta}^{\top} \boldsymbol{x})$, for $\boldsymbol{x}$ either a test or

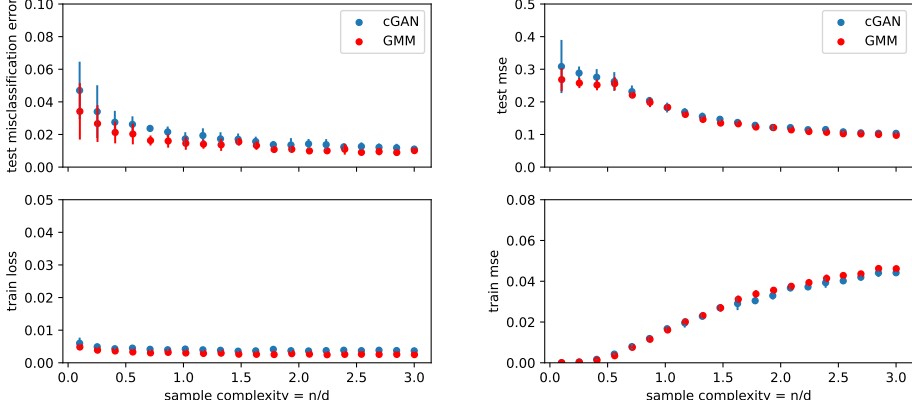

Figure 2: Illustration of the universality scenario described in Fig.1. Logistic (left) & ridge (right) regression test (up) and training (bottom) errors are shown versus the sample complexity $\alpha = n/d$ for an odd vs. even binary classification task on two data models: Blue dots data generated from a conditional GAN [41] trained on the fashion-MNIST dataset [45] and pre-processed with a random features map $\boldsymbol{x} \mapsto \tanh(W\boldsymbol{x})$ with Gaussian weights $W \in \mathbb{R}^{1176 \times 784}$; Red dots are the 10- clusters Gaussian mixture model with means and covariances matching each fashion-MNIST cluster conditioned on labels ($\ell_2$ regularization is $\lambda = 10^{-4}$). Details on the simulations are discussed in Appendix D.

training sample from (1). For instance, in a Gaussian mixture model, where $\boldsymbol{x} \sim \sum_{c \in \mathcal{C}} \rho_c \mathcal{N}(\boldsymbol{\mu}_c, \boldsymbol{\Sigma}_c)$, the sufficient statistics are simply given by the first two moments of these pre-activations. However, for a general mixture model (1), the sufficient statistics will generically depend on all moments of these pre-activations. Surprisingly, our key result in this work is to show that in the high-dimensional limit this is not the case. In other words, under some conditions which are made precise in Section 2, we show that expectations with respect to (1) can be exchanged by expectations over a Gaussian mixture with matching moments. This can be formalized as follows. Define an *equivalent Gaussian data set* $(\boldsymbol{g}_i, y_i)_{i=1}^n \in \mathbb{R}^p \times \mathcal{Y}$ with samples drawn *i.i.d.* from the *equivalent Gaussian mixture model*:

$$\boldsymbol{g}_i \sim \sum_{c \in \mathcal{C}} \rho_c \mathcal{N}(\boldsymbol{\mu}_c, \boldsymbol{\Sigma}_c), \qquad\qquad y_i(\boldsymbol{G}) = \eta(\boldsymbol{\Theta}_\star^\top \boldsymbol{g}_i, \varepsilon_i, c_i). \qquad (8)$$

We recall that $\boldsymbol{\mu}_c, \boldsymbol{\Sigma}_c$ denotes the mean and covariance of $P_c^{\boldsymbol{x}}$ from (1). Consider a family of estimators $(\boldsymbol{\Theta}_1, \cdots, \boldsymbol{\Theta}_M)$ defined by minimization (3) and/or sampling (5) over the training data $(\boldsymbol{X}, \boldsymbol{y})$ from the mixture model (1). Let $h$ be a statistical metric of interest. Then, in the proportional high-dimensional limit where $n, p \to \infty$ at a fixed $\alpha = n/d > 0$, and where $\langle \cdot \rangle$ denote the expectation with respect to the Gibbs distribution (5), we define universality as:

$$\mathbb{E}_{\boldsymbol{X}} \left[ \langle h(\boldsymbol{\Theta}_1, \cdots, \boldsymbol{\Theta}_M) \rangle_{\boldsymbol{X}} \right] \underset{n \to \infty}{\simeq} \mathbb{E}_{\boldsymbol{G}} \left[ \langle h(\boldsymbol{\Theta}_1, \cdots, \boldsymbol{\Theta}_M) \rangle_{\boldsymbol{G}} \right] \qquad (9)$$

The goal of the next section is to make this statement precise.

## 2   Main results

We now present the main theoretical contributions of the present work and discuss its consequences. Our proofs for Theorems 4 and 6 build upon existing results on the universality of empirical risk minimization for uni-model distributions [16, 17] and therefore rely on similar technical regularity and concentration assumptions. Concretely, our work relies on the following assumptions:

**Assumption 1** (Loss and regularization)**.** *The loss function $\ell : \mathbb{R}^{k+1} \to \mathbb{R}$ is nonnegative and Lipschitz, and the regularization function $r : \mathbb{R}^{p \times k} \to \mathbb{R}$ is locally Lipschitz, with constants independent from $p$.*

**Assumption 2** (Boundedness and concentration). *The constraint set $\mathcal{S}_p$ is a compact subset of $\mathbb{R}^{k \times p}$. Further, there exists a constant $M > 0$ such that for any $c \geq 0$,*

$$\sup_{\boldsymbol{\theta} \in \mathcal{K}_p, \|\boldsymbol{\theta}\|_2 \leq 1} \|\boldsymbol{\theta}^\top \boldsymbol{x}\|_{\psi_2} \leq M, \quad \sup_{\boldsymbol{\theta} \in \mathcal{K}_p, \|\boldsymbol{\theta}\|_2 \leq 1} \|\boldsymbol{\Sigma}_c^{1/2} \boldsymbol{\theta}\|_2 \leq M, \quad and \quad \|\boldsymbol{\mu}_c\|_2 \leq M \tag{10}$$

*where $\|\cdot\|_{\psi_2}$ is the sub-gaussian norm, and $\mathcal{K}_p \subseteq \mathbb{R}^p$ is such that $\mathcal{S}_p \subseteq \mathcal{K}_p^k$.*

**Assumption 3** (Labels). *The labeling function $\eta$ is Lipschitz, the teacher vector $\boldsymbol{\Theta}$ belongs to $\mathcal{S}_p$, and the noise variables $\varepsilon_i$ are i.i.d sub-gaussian with $\|\varepsilon_i\|_{\psi_2} \leq M$ for some constant $M > 0$.*

Those three assumptions are fairly technical, and it is possible that the universality properties proven in this article hold irrespective of these conditions. The crucial assumption in our theorems is that of a *conditional one-dimensional CLT*:

**Assumption 4.** *For any Lipschitz function $\varphi : \mathbb{R} \to \mathbb{R}$,*

$$\lim_{n,p \to \infty} \sup_{\boldsymbol{\theta} \in \mathcal{K}_p} \left| \mathbb{E}\left[\varphi(\boldsymbol{\theta}^\top \boldsymbol{x}) \, \middle| \, c_{\boldsymbol{x}} = c\right] - \mathbb{E}\left[\varphi(\boldsymbol{\theta}^\top \boldsymbol{g}) \, \middle| \, c_{\boldsymbol{g}} = c\right] \right| = 0, \quad \forall c \in \mathcal{C} \tag{11}$$

where $\boldsymbol{x}$ and $\boldsymbol{g}$ denote samples from the given mixture distribution and the equivalent gaussian mixture distribution in equations (1) and (8) respectively.

The above assumption is a generalization of the "one-dimensional CLT" underlying the line of work based on the Gaussian equivalence (GE) Principle [13, 14, 17, 16]. The above assumption splits the proof of universality for a general mixture distribution into two parts. First, one shows that asymptotic universality of an observable $h$ can be reduced to the proof of a one-dimensional CLT. Second, one proves this CLT holds for the particular class of features of interest. This proof scheme streamlines universality proofs. Our work provides a general proof of the first step in Theorem 4, conditioned on the second, later showing that Assumption 4 holds for some natural feature maps of interest, i.e. random features applied to a Gaussian mixture (Theorem 6). However, Assumption 4 is expected to hold for a large class of features, as supported by our empirical observations in Figure 2 and arguments in Appendix C.

## 2.1 Universality of Mixture Models

We start by proving the universality of the free energy for a Gibbs distribution defined through the objective $\widehat{\mathcal{R}}_n(\boldsymbol{\Theta}; \boldsymbol{X}, \boldsymbol{y})$ for the data distribution (1) and its equivalent Gaussian mixture (8).

**Theorem 1** (Universality of Free Energy). *Let $\mu_p(\boldsymbol{\Theta})$ be a sequence of Borel probability measures with compact supports $\mathcal{S}_p$. Define the following free energy function:*

$$f_{\beta,n}(\boldsymbol{X}) = -\frac{1}{\beta n} \log \int \exp\left(-\beta n \widehat{\mathcal{R}}_n(\boldsymbol{\Theta}; \boldsymbol{X}, \boldsymbol{y}(\boldsymbol{X}))\right) d\mu_p(\boldsymbol{\Theta}) \tag{12}$$

*Under Assumptions 1-4 on $\boldsymbol{X}$ and $\mathcal{S}_p$, for any bounded differentiable function $\Phi$ with bounded Lipschitz derivative, we have:*

$$\lim_{n,p \to \infty} \left| \mathbb{E}\left[\Phi\left(f_{\beta,n}(\boldsymbol{X})\right)\right] - \mathbb{E}\left[\Phi\left(f_{\beta,n}(\boldsymbol{G})\right)\right] \right| = 0.$$

When $\mu_p$ corresponds to discrete measures supported on an $\epsilon$-net in $\mathcal{S}_p$, using the reduction from Lemma 1 to Theorem 1 in [17], we obtain the following corollary:

**Corollary 2** (Universality of Training Error). *For any bounded Lipschitz function $\Phi : \mathbb{R} \to \mathbb{R}$:*

$$\lim_{n,p \to \infty} \left| \mathbb{E}\left[\Phi\left(\widehat{\mathcal{R}}_n^\star(\boldsymbol{X}, \boldsymbol{y}(\boldsymbol{X}))\right)\right] - \mathbb{E}\left[\Phi\left(\widehat{\mathcal{R}}_n^\star(\boldsymbol{G}, \boldsymbol{y}(\boldsymbol{G}))\right)\right] \right| = 0$$

*In particular, for any $\mathcal{E} \in \mathbb{R}$, and denoting $\xrightarrow{\mathbb{P}}$ the convergence in probability:*

$$\widehat{\mathcal{R}}_n^\star(\boldsymbol{X}, \boldsymbol{y}(\boldsymbol{X})) \xrightarrow{\mathbb{P}} \mathcal{E} \quad if \ and \ only \ if \quad \widehat{\mathcal{R}}_n^\star(\boldsymbol{G}, \boldsymbol{y}(\boldsymbol{G})) \xrightarrow{\mathbb{P}} \mathcal{E}, \tag{13}$$

The full theorem, as well as its proof, is presented in Appendix A, along with additional remarks and an intuitive sketch. The proof combines the conditional 1d-CLT in Assumption 4 with the

interpolation of the free-energy in [17]. For strongly-convex losses, one may alternatively use the Lindeberg's method as in [16].

In a nutshell, this theorem shows that the multi-modal data generated by any generative neural network is equivalent to a *finite* mixture of Gaussian in high-dimensions: in other words, a *finite* mixture of Gaussians leads to the same loss as for data generated by (for instance) a cGAN. Since the function $\ell : \mathbb{R}^{k+1} \to \mathbb{R}$ need not be convex, we can take

$$\ell(\boldsymbol{x}_{\text{out}}, y) = \ell'(\boldsymbol{\Psi}(\boldsymbol{x}_{\text{out}}), y),$$

where $\boldsymbol{\Psi}$ is an already pretrained neural network. In particular, if $\boldsymbol{x}$ is the output of all but the last layer of a neural net, we can view $\boldsymbol{\Psi}$ as the averaging procedure for a small committee machine.

Note that Corollary 2 depends crucially on Assumption 4 (the one-dimensional CLT), which is by no means evident. We discuss the conditions on the weights matrix for which it can be proven in Section 2.4. However, one can observe empirically that the validity of Corollary 2 goes well beyond what can be currently proven. A number of numerical illustrations of this property can be found in the work of [23, 39, 40], who already derived similar (albeit more limited) results using random matrix theory. Additionally, we observed that even with trained GANs, when we observed data through a random feature map [46], the Gaussian mixture universality is well obeyed. This scenario is illustrated in Fig. 1, with a concrete example in Fig. 2. Even though we did not prove the one-dimensional CLT for arbitrary learned matrices, and worked with finite moderate sizes, the realistic data generated by our cGAN behaves extremely closely to those generated by the corresponding Gaussian mixture.

A second remark is that the interest of Corollary 2 lies in the fact that it requires only a *finite* mixture to approximate the loss. Indeed, while we could use the standard approximation results (e.g. the Stone-Weierstrass theorem) to approximate the data density to arbitrary precision by Gaussian mixtures, this would require a diverging number of Gaussian in the mixture. The fact that loss is captured with finite $\mathcal{C}$ is key to our approach.

## 2.2 Convergence of expectations for Joint Minimization and Sampling

Our next result establishes a general relationship between the differentiability of the limit of expected training errors or free energies for empirical risk minimization or free energies for sampling and the universality of expectations of a given function of a set of parameters arising from multiple objectives. As a motivating example, consider the uncertainty quantification in Section 1 that uses both Bayesian and ERM estimators [19, 24]. The parameters $\hat{\boldsymbol{\Theta}}_{\text{erm}}$ and $\boldsymbol{\Theta}_{\text{Bayes}}$ are obtained through empirical risk minimization and posterior sampling respectively on the same sequence of training data. In general, the inputs used in different objectives could be different but have some correlation structure. In the setup of ensembling (Equation 7), they are correlated through the feature mapping $\boldsymbol{z}_i \mapsto \boldsymbol{x}_i^{(m)} = \varphi_m(\boldsymbol{z}_i)$. In light of these considerations, we present the following general setup: Consider a sequence of $M$ risks:

$$\widehat{\mathcal{R}}_n^{(m)}(\boldsymbol{\Theta}; \boldsymbol{X}^{(m)}, \boldsymbol{y}^{(m)}) := \frac{1}{n} \sum_{i=1}^{n} \ell_m(\boldsymbol{\Theta}^\top \boldsymbol{x}_i^{(m)}, y_i^{(m)}) + r_m(\boldsymbol{\Theta}), \quad m \in [M] \tag{14}$$

with possibly different losses, regularizers and datasets. For simplicity, we assume that the objectives are defined on parameters having the same dimension $\boldsymbol{\Theta} \in \mathbb{R}^{p \times k}$. We aim to minimize $M_1$ of them:

$$\hat{\boldsymbol{\Theta}}^{(m)}(\boldsymbol{X}) \in \underset{\boldsymbol{\Theta} \in \mathcal{S}_p^{(m)}}{\arg\min} \; \widehat{\mathcal{R}}_n^{(m)}(\boldsymbol{\Theta}; \boldsymbol{X}^{(m)}, \boldsymbol{y}^{(m)}), \quad m \in [M_1] \tag{15}$$

and the $M_2$ remaining parameters are independently sampled from a family of Gibbs distributions:

$$\boldsymbol{\Theta}^{(m)} \sim P_m(\boldsymbol{\Theta}) \propto \exp\left(-\beta_m \widehat{\mathcal{R}}_n^{(m)}\left(\boldsymbol{\Theta}; \boldsymbol{X}^{(m)}, \boldsymbol{y}^{(m)}\right)\right) d\mu_m(\boldsymbol{\Theta}), \quad m \in [M_1 + 1, M], \tag{16}$$

where $M = M_1 + M_2$. The joint distribution of the $\boldsymbol{x}_i = (\boldsymbol{x}_i^{(1)}, \dots, \boldsymbol{x}_i^{(M)})$ is assumed to be a mixture of the form (1). However, we assume that the labels $y_i^{(m)}$ only depend on the vectors $\boldsymbol{x}_i^{(m)}$:

$$y_i^{(m)}(\boldsymbol{X}^{(m)}) = \eta(\boldsymbol{\Theta}_\star^{(m)\top} \boldsymbol{x}_i^{(m)}, \varepsilon_i^{(m)}, c_i). \tag{17}$$

The equivalent Gaussian inputs $\boldsymbol{g}_i = (\boldsymbol{g}_i^{(1)}, \dots, \boldsymbol{g}_i^{(M)})$ and their labels $\boldsymbol{y}(\boldsymbol{G})$ are defined as in (8).

**Statistical metric and free energy —** Our goal is to study statistical metrics for some function $h : \mathbb{R}^{M \times k \times p} \to \mathbb{R}$ of the form $h(\boldsymbol{\Theta}^{(1)}, \cdots, \boldsymbol{\Theta}^{(M)})$. For instance, the metric $h$ could be the population risk (a.k.a. generalization error), or some overlap between $\boldsymbol{\Theta}$ and $\boldsymbol{\Theta}_\star$. We define the following coupling free energy function:

$$f_{n,s}(\boldsymbol{\Theta}[1:M_1], \boldsymbol{X}, \boldsymbol{y}) = -\frac{1}{n}\log \int e^{-sn\, h(\boldsymbol{\Theta}^{(1)}, \dots, \boldsymbol{\Theta}^{(M)})} dP^{(M_1+1):M}, \tag{18}$$

where $P^{(M_1+1):M}$ denotes the product measure of the $P_m$ defined in (16). This gives rise to the following joint objective:

$$\widehat{\mathcal{R}}_{n,s}(\boldsymbol{\Theta}[1:M_1], \boldsymbol{X}, \boldsymbol{y}) = \sum_{m=1}^{M_1} \widehat{\mathcal{R}}_n^{(m)}(\boldsymbol{\Theta}^{(m)}; \boldsymbol{X}^{(m)}, \boldsymbol{y}^{(m)}) + f_{n,s}(\boldsymbol{\Theta}[1:M_1], \boldsymbol{X}, \boldsymbol{y}). \tag{19}$$

In particular, when $s = 0$ we have $f_{n,0} = 0$ and the problem reduces to the joint minimization problem in (15). Our first result concerns the universality of the minimum of the above problem:

**Proposition 3** (Universality for joint minimization and sampling)**.** *Under Assumptions 1-4, for any $s > 0$ and bounded Lipschitz function $\Phi : \mathbb{R} \to \mathbb{R}$, and denoting $\widehat{\mathcal{R}}_{n,s}^\star(\boldsymbol{X}, \boldsymbol{y}) \coloneqq \min \widehat{\mathcal{R}}_{n,s}(\boldsymbol{\Theta}; \boldsymbol{X}, \boldsymbol{y})$:*

$$\lim_{n,p\to\infty} \left| \mathbb{E}\left[ \Phi\left( \widehat{\mathcal{R}}_{n,s}^\star(\boldsymbol{X}, \boldsymbol{y}(\boldsymbol{X})) \right) \right] - \mathbb{E}\left[ \Phi\left( \widehat{\mathcal{R}}_{n,s}^\star(\boldsymbol{G}, \boldsymbol{y}(\boldsymbol{G})) \right) \right] \right| = 0$$

The proof uses a reduction to Corollary 2, and can be found in App. A.5. The next result concerns the value of $h$ at the minimizers point $(\hat{\boldsymbol{\Theta}}^{(1)}, \dots, \hat{\boldsymbol{\Theta}}^{(M)})$. We make the following additional assumptions:

**Assumption 5** (Differentiable Limit)**.** *There exists a neighborhood of $0$ such that the function $q_n(s) = \mathbb{E}\left[\widehat{\mathcal{R}}_{n,s}^\star(\boldsymbol{G}, \boldsymbol{y}(\boldsymbol{G}))\right]$ converges pointwise to a function $q(s)$ that is differentiable at $0$.*

The above assumption stems from the convexity based argument used to prove Theorem 4.

For a fixed realization of the dataset $\boldsymbol{X}$, we denote by $\left\langle h(\boldsymbol{\Theta}^{(1)}, \cdots, \boldsymbol{\Theta}^{(M)}) \right\rangle_{\boldsymbol{X}}$ the expected value of $h$ when $(\hat{\boldsymbol{\Theta}}^{(1)}, \dots, \hat{\boldsymbol{\Theta}}^{(M_1)})$ are obtained through the minimization of (15) and $(\boldsymbol{\Theta}^{(M_1+1)}, \dots, \boldsymbol{\Theta}^{(M)})$ are sampled according to the Boltzmann distributions (16).

**Assumption 6.** *With high probability on $\boldsymbol{X}, \boldsymbol{G}$, the value $\left\langle h(\boldsymbol{\Theta}^{(1)}, \cdots, \boldsymbol{\Theta}^{(M)}) \right\rangle_{\boldsymbol{X}}$ (resp. the same for $\boldsymbol{G}$) is independent from the chosen minimizers in (15).*

The above assumption is motivated by the fact that commonly non-convex problems contain minima exhibiting a specific symmetry. For example, all the global minima for a two-layer neural network are permutation invariant. Assumption 6 reflects that the quantity $h$ respects these symmetries by taking the same value at each global minimum. This can be replaced by the stronger condition of a unique minimizer. Then the following holds:

**Theorem 4.** *Under Assumptions 1-6, we have:*

$$\lim_{n,p\to\infty} \left| \mathbb{E}\left[ \left\langle h\left( \boldsymbol{\Theta}^{(1)}, \cdots, \boldsymbol{\Theta}^{(M)} \right) \right\rangle_{\boldsymbol{X}} \right] - \mathbb{E}\left[ \left\langle h\left( \boldsymbol{\Theta}^{(1)}, \cdots, \boldsymbol{\Theta}^{(M)} \right) \right\rangle_{\boldsymbol{G}} \right] \right| = 0, \tag{20}$$

**Proof Sketch:** Our proof relies on the observation that $q_n(s)$ is a concave function of $s$. Further:

$$q_n'(0) = \mathbb{E}\left[ \left\langle h\left( \boldsymbol{\Theta}^{(1)}, \cdots, \boldsymbol{\Theta}^{(M)} \right) \right\rangle_{\boldsymbol{G}} \right]. \tag{21}$$

This allows us to leverage a result of convex analysis relating the convergence of a sequence of convex or concave functions to the convergence of the corresponding derivatives, bypassing the more involved probabilistic arguments in [16, 17]. Our approach also generalizes in a straightforward manner to the setup of multiple objectives.

The above result shows that the expected value of $h\left( \boldsymbol{\Theta}^{(1)}, \cdots, \boldsymbol{\Theta}^{(M)} \right)$ for a multi-modal data satisfying the 1d CLT is equivalent to that of a mixture of Gaussians. The full theorem is presented and proven in Appendix A.

## 2.3 Universal Weak Convergence

Theorem 4 provides a general framework for proving the equivalence of arbitrary functions of parameters obtained by minimization/sampling on a given mixture dataset and the equivalent gaussian mixture distribution. However, it relies on the assumption of a differentiable limit of the free energy (Assumption 5). If the assumption holds for a sequence of functions belonging to dense subsets of particular classes of functions, it allows us to prove convergence of minimizers themselves, in a weak sense. We illustrate this through a simple setup considered in [12], which precisely characterized the asymptotic distribution of the minimizers of empirical risk with GMM data in the strictly convex case. Consider the following setup:

$$\left(\hat{\boldsymbol{W}}^{\boldsymbol{X}}, \hat{\boldsymbol{b}}^{\boldsymbol{X}}\right) = \underset{\boldsymbol{W}, \boldsymbol{b}}{\arg\min} \sum_{i=1}^{n} \ell\left(\frac{\boldsymbol{W}\boldsymbol{x}_i}{\sqrt{d}} + \boldsymbol{b}, \boldsymbol{y}_i\right) + \lambda r(\boldsymbol{W}), \tag{22}$$

where $\boldsymbol{W} \in \mathbb{R}^{|\mathcal{C}|\times d}$, $\boldsymbol{b} \in \mathbb{R}^{|\mathcal{C}|}$ and $\boldsymbol{y}_i \in \mathbb{R}^{|\mathcal{C}|}$ is the one-hot encoding of the class index $c_i$. We make the following assumptions:

**Assumption 7.** *All of the covariance matrices $\Sigma_c$ are diagonal, with strictly positive eigenvalues $(\sigma_{c,i})_{i\in[d]}$, and there exists a constant $M > 0$ such that for any $c \in \mathcal{C}$ we have $\sigma_{c,i} \leq M$ and $\|\boldsymbol{\mu}_c\|_2 \leq M$.*

Secondly, since we aim at obtaining a result on the weak convergence of the estimators, we assume the same weak convergence for the means and covariances, and that the regularization only depends on the empirical measure of $\boldsymbol{W}$.

**Assumption 8.** *The empirical distribution of the $\boldsymbol{\mu}_c$ and $\Sigma_c$ converges weakly as follows:*

$$\frac{1}{d}\sum_{i=1}^{d}\prod_{c\in\mathcal{C}}\delta(\mu_c - \sqrt{d}\mu_{c,i})\delta(\sigma_c - \sigma_{c,i}) \xrightarrow[d\to\infty]{\mathcal{L}} p(\boldsymbol{\sigma}, \boldsymbol{\mu}) \tag{23}$$

**Assumption 9.** *The regularizer $r(\cdot)$ is a pseudo-Lipshitz function of finite-order having the following form: $r(\boldsymbol{W}) = \sum_{i=1}^{d}\psi_r(\boldsymbol{W}_i)$, for some convex, differentiable function $\psi_r : \mathbb{R} \to \mathbb{R}$. This includes, in particular the squared regularization $r(\boldsymbol{W}) = \sum_{i=1}^{d}\boldsymbol{W}_i^2$.*

We briefly comment on the choice of the above assumptions. The boundedness of the $\Sigma_c$ and $\mu$ in Assumption 7 guarantees that we are in a case covered both by [12] and by the assumptions of Theorem 4. The diagonal property of the $\Sigma_c$ in 7, as well as the joint convergence in Assumption 8, ensure that we can view the minimization problem 22 ensures that $W^\star, b^\star$ converge towards a well-defined limit. Finally, the separability assumption on $r$ in assumption 9 responds to the fact that we aim for a result on the empirical coordinate distribution of $W^\star, b^\star$

Under these conditions, the joint empirical measure of the minimizers and of the data moments converges weakly to a fixed limit, independent of the data-distribution:

**Theorem 5.** *Assume that conditions 1-9 hold, and further that the function $\ell(\bullet, y) + r(\bullet)$ is convex, coercive and differentiable. Then, for any bounded-Lipschitz function: $\Phi : \mathbb{R}^{3|\mathcal{C}|} \to \mathbb{R}$, we have:*

$$\mathbb{E}\left[\frac{1}{d}\sum_{i=1}^{d}\Phi(\{(\hat{\boldsymbol{W}}^{\boldsymbol{X}})_{c,i}\}_{c\in\mathcal{C}}, \{\mu_{c,i}\}_{c\in\mathcal{C}}, \{\sigma_{c,i}\}_{c\in\mathcal{C}})\right] \xrightarrow[n/d=\alpha>0]{n,d\to+\infty} \mathbb{E}_{\tilde{p}}\left[\Phi(\boldsymbol{w}, \boldsymbol{\mu}, \boldsymbol{\sigma})\right], \tag{24}$$

*where $\tilde{p}$ is a measure on $\mathbb{R}^{3|\mathcal{C}|}$, that is determined by the so-called* replica equations.

**Proof Sketch** The proof starts with the observation that the nonlinear system of (replica) equations in [12] describes the joint-empirical measure of the parameters, means and covariances of the mixtures in a self-consistent manner. Furthermore, for $h(\boldsymbol{W})$ having bounded second derivatives, the perturbation term $sh(\boldsymbol{W})$ can be absorbed into the regularizer. We then utilize topological and analytical arguments to relate the weak convergence to the differentiability Assumption 5 for functions that can be expressed as expectations w.r.t the joint empirical measure in 24. More details can be found in Appendix A.8.

In particular, the above result implies the universality of the overlaps of the minimizers with means, covariances, as well as their geometrical properties such as $L^p$ norms.

## 2.4 One-dimensional CLT for Random Features

We finally show a conditional one-dimensional CLT for a random features map applied to a mixture of gaussians, in the vein of those shown in [14, 16, 17]. Concretely, we consider the following setup:

$$\boldsymbol{x}_i = \sigma(\boldsymbol{F}\boldsymbol{z}_i), \quad \boldsymbol{z}_i \sim \sum_{c \in \mathcal{C}} \mathcal{N}(\boldsymbol{\mu}_c^{\boldsymbol{z}}, \boldsymbol{\Sigma}_c^{\boldsymbol{z}}), \tag{25}$$

where the feature matrix $\boldsymbol{F} \in \mathbb{R}^{p \times d}$ has i.i.d $\mathcal{N}(0, 1/d)$ entries. This setup is much more permissive than the ones in [16, 17], that restrict the samples $\boldsymbol{z}$ to standard normal vectors. However, we do require some technical assumptions:

**Assumption 10.** *The activation function $\sigma$ is thrice differentiable, with $\|\sigma^{(i)}\| \le M$ for some $M > 0$, and we have $\mathbb{E}_{g \sim \mathcal{N}(0,1)}[\sigma(g)] = 0$. Additionally, the cluster means and covariances of $\boldsymbol{z}$ satisfy for all $c \in \mathcal{C}$ $\|\boldsymbol{\mu}_c^{\boldsymbol{z}}\| \le M, \|\boldsymbol{\Sigma}_c^{\boldsymbol{z}}\|_{\mathrm{op}} \le M$ for some constant $M > 0$.*

We also place ourselves in the proportional regime, i.e. a regime where $p/d \in [\gamma^{-1}, \gamma]$ for some $\gamma > 0$. For simplicity, we will consider the case $k = 1$; and the constraint set $\mathcal{S}_p$ as follows:

$$\mathcal{S}_p = \left\{ \boldsymbol{\theta} \in \mathbb{R}^d \,\middle|\, \|\boldsymbol{\theta}\|_2 \le R, \quad \|\boldsymbol{\theta}\|_\infty \le Cp^{-\eta} \right\} \tag{26}$$

for a given $\eta > 0$. We show in the appendix the following theorem:

**Theorem 6.** *Under Assumption 10, and with high probability on the feature matrix $\boldsymbol{F}$, the data $\boldsymbol{X}$ satisfy the concentration assumption 2, as well as the one-dimensional CLT of Assumption 4. Consequently, the results of Theorems 1 and 4 apply to $\boldsymbol{X}$ and their Gaussian equivalent $\boldsymbol{G}$.*

**Proof Sketch**    Our proof proceeds by defining the following neuron-wise activation functions:

$$\sigma_{i,c}(u) = \sigma(u + \boldsymbol{f}_i^\top \boldsymbol{\mu}_c). \tag{27}$$

We subsequently control the effects of the means, covariances and the dimensions of the inputs to prove a result analogous to the one-dimensional CLT for random features in [16, 17, 14]. While we prove the above result for random weights, we note, however that the non-asymptotic results in [16, 14] also hold for deterministic matrices satisfying approximate orthogonality conditions. Therefore, we expect the one-dimensional CLT to approximately hold for a much larger class of feature maps. Finally, we also note that the above extension of the one-dimensional CLT to mixture of gaussians also provides a proof for the asymptotic error for random features in [11].

**Conclusions —** We demonstrate the universality of the Gaussian mixture assumption in high-dimension for various machine learning tasks such as empirical risk minimization, sampling and ensembling, in a variety of settings including random features or GAN generated data. We also show that universality holds for a large class of functions, and provide a weak convergence theorem. These results, we believe, vindicate the classical theoretical line of works on the Gaussian mixture design. We hope that our results will stimulate further research in this area. We also believe it crucial to understand the limitations of our extended universality framework, for instance in the cases of data with low-dimensional structure or sparsity.

# Acknowledgements

We acknowledge funding from the ERC under the European Union's Horizon 2020 Research and Innovation Program Grant Agreement 714608-SMiLe, as well as by the Swiss National Science Foundation grant SNFS OperaGOST, 200021_200390.

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
