# A    Proofs of Main Results

## A.1    Notation

We follow the setting defined in section 1. Throughout, we work in the so-called proportional high-dimensional limit, where $n, p$ go to infinity with

$$\frac{n}{p} \to \alpha > 0,$$

while $\mathcal{C}$ stays fixed.

Throughout this section, $\| \cdot \|$ will denote the spectral norm of a matrix, while $\| \cdot \|_q$ for $q > 0$ will refer to the element-wise $q$-norms. For a subgaussian random variable $Y$, its sub-gaussian norm $\|Y\|_{\psi_2}$ is defined as

$$\|Y\|_{\psi_2} = \inf\left\{ t > 0 \,\middle|\, \mathbb{E}\left[\exp\left(\frac{Y^2}{t^2}\right)\right] \leq 2\right\}.$$

## A.2    State of the art

**Proof sketch and remarks —**    Before providing a complete proofs, we believe it is useful to present a short intuitive presentation explaining why these results hold.

(i) A crucial first remark is that Theorem 1 and Corollary 2 do not require that the data generated by GANs are Gaussians mixtures: rather, it is their one-dimensional projections along the directions in $\mathcal{S}_p$ that should behave as such. The first, intuitive, explanation is that indeed in high dimension, for a randomly chosen vector $\boldsymbol{\theta} \in \mathcal{S}_p$, it is natural to expect that $\boldsymbol{\theta}^\top \boldsymbol{x})$ behaves like a Gaussian mixture. Indeed, if we condition on a given label, then $z$ is Gaussian, the random variable $\boldsymbol{x} = \Psi_{\mathrm{nn}}(\mathbf{z})$, has a well defined mean and variance (at least if $\Psi$ is a Lipschitz function), so that the central limit theorem shows that $\boldsymbol{\theta}^\top \boldsymbol{x})$ converges to a Gaussian variable.

(ii) We require, however, a slightly stronger condition in eq. (11): Indeed, it should be that, conditioned on a label, $\boldsymbol{\theta} \in \mathcal{S}_p$ is Gaussian *for all* $\boldsymbol{\theta} \in \mathcal{S}_p$, not a randomly chosen one (since we do certainly do not chose our weights randomly). This condition might appear strong. However, such one-dimensional CLTs have been the subject of many recent works which proved them for many cases [14, 16, 17], including random features and two-layers neural tangent kernels. We extend the proof of the one-dimensional CLT to mixture models in section 2.4. We also provide further formal arguments in App. C. In particular, we argue that a large class of distributions, including deep generative models, do satisfy this condition that can also be checked empirically in simulations [14, 40].

(iii) The one-dimensional CLT now implies that $\mathbb{E}\left[\widehat{\mathcal{R}}_n\left(\boldsymbol{\Theta}; \boldsymbol{X}, \boldsymbol{y}(\boldsymbol{X})\right)\right] \simeq \mathbb{E}\left[\widehat{\mathcal{R}}_n\left(\boldsymbol{\Theta}; \boldsymbol{G}, \boldsymbol{y}(\boldsymbol{G})\right)\right]$ for any **fixed** choice of $\boldsymbol{\Theta}$, **independent from the data**. There is another, additional difficulty: when one performs empirical risk minimization, the minimizer $\hat{\boldsymbol{\Theta}}(\boldsymbol{X})$ **strongly depends** on the data $\boldsymbol{X}$, and the naive 1d-CLT simply does not apply! Solving the problem of the dependence of the estimator over the data is the main mathematical difficulty in proving Thm. 2. This is achieved in the Appendix by using a method due to [17], that leverage on the Guerra interpolation techniques used to prove the validity of the replica method [47]. The idea is to define a t-dependent model that uses $n$ dataset points at $t = 0$, and GMM ones at $t = 1$, and to show that the free energy (and all observables) remains constants at all "times" $t \in [0, 1]$). This establishes fully the universality advocated in our theorem.                                                                                $\square$

We now review the recent result of [17]. Consider the minimization problem (4), with $(\boldsymbol{x}_\mu, y_\mu)$ i.i.d random variables; the goal is to replace the $\boldsymbol{x}_\mu$ by their Gaussian equivalent model

$$\boldsymbol{g}_i \overset{i.i.d}{\sim} \mathcal{N}(\boldsymbol{\mu}, \boldsymbol{\Sigma}) \qquad \text{where} \qquad \boldsymbol{\mu} = \mathbb{E}\left[\boldsymbol{x}\right], \quad \boldsymbol{\Sigma} = \mathbb{E}\left[\boldsymbol{x}\boldsymbol{x}^\top\right]. \tag{28}$$

[17] make the following assumptions:

**Assumption A1** (Loss and regularization). *The loss function $\ell : \mathbb{R}^{k+1} \to \mathbb{R}$ is nonnegative and Lipschitz, and the regularization function $r : \mathbb{R}^{p \times k} \to \mathbb{R}$ is locally Lipschitz, with constants independent from $p$.*

**Assumption A2** (Concentration on the directions of $\mathcal{S}_p$). *We have*

$$\sup_{\boldsymbol{\theta} \in \mathcal{S}_p, \|\boldsymbol{\theta}\|_2 \leq 1} \|\boldsymbol{\theta}^\top \boldsymbol{x}\|_{\psi_2} \leq M, \qquad \sup_{\boldsymbol{\theta} \in \mathcal{S}_p, \|\boldsymbol{\theta}\|_2 \leq 1} \|\boldsymbol{\Sigma}^{1/2}\boldsymbol{\theta}\|_2 \leq M, \quad \text{and} \quad \|\boldsymbol{\mu}\|_2 \leq M \qquad (29)$$

*for some constant $M > 0$.*

**Assumption A3** (One-dimensional CLT). *For any bounded Lipschitz function $\varphi : \mathbb{R}^k \to \mathbb{R}$,*

$$\lim_{p \to \infty} \sup_{\boldsymbol{\theta} \in \mathcal{S}_p} \left| \mathbb{E} \left[ \phi(\boldsymbol{\theta}^\top \boldsymbol{x}) \right] - \mathbb{E} \left[ \phi(\boldsymbol{\theta}^\top \boldsymbol{g}) \right] \right| \big| = 0. \qquad (30)$$

**Assumption A4** (Labels). *The $y_\mu$ are generated according to*

$$y_i = \eta(\boldsymbol{\Theta}^* \boldsymbol{x}_i, \varepsilon_i, c_i), \qquad (31)$$

*where $\eta : \mathbb{R}^{k^*+1} \to \mathbb{R}$ is a Lipschitz function, $\boldsymbol{\Theta}^* \in \mathcal{S}_p^{k^*}$, and the $\varepsilon_\mu$ are i.i.d subgaussian random variables with*

$$\|\varepsilon_i\|_{\psi_2} \leq M$$

*for some constant $M > 0$.*

Building on those assumptions, [17] prove the following:

**Theorem 7** (Theorem 1. in [17]). *Suppose that Assumptions A1-A3 hold. Then, for any bounded Lipschitz function $\Phi : \mathbb{R} \to \mathbb{R}$, we have*

$$\lim_{n,p \to \infty} \left| \mathbb{E} \left[ \Phi \left( \widehat{\mathcal{R}}_n^\star(\boldsymbol{X}, \boldsymbol{y}(\boldsymbol{X})) \right) \right] - \mathbb{E} \left[ \Phi \left( \widehat{\mathcal{R}}_n^\star(\boldsymbol{G}, \boldsymbol{y}(\boldsymbol{G})) \right) \right] \right| = 0$$

*In particular, for any $\rho \in \mathbb{R}$,*

$$\widehat{\mathcal{R}}_n^\star(\boldsymbol{X}, \boldsymbol{y}(\boldsymbol{X})) \xrightarrow{\mathbb{P}} \rho \quad \text{if and only if} \quad \widehat{\mathcal{R}}_n^\star(\boldsymbol{G}, \boldsymbol{y}(\boldsymbol{G})) \xrightarrow{\mathbb{P}} \rho$$

**Free energy approximation** A crucial component of the proof in [17] is the approximation of the minimizer through a free energy function. Define the discretized free energy

$$f_{\epsilon,\beta}(\boldsymbol{X}) = -\frac{1}{n\beta} \sum_{\boldsymbol{\Theta} \in \mathcal{N}_\epsilon^k} \exp \left( -\beta \, \widehat{\mathcal{R}}_n(\boldsymbol{\Theta}; \boldsymbol{X}, \boldsymbol{y}(\boldsymbol{X})) \right), \qquad (32)$$

where $\mathcal{N}_\epsilon$ is a minimal $\epsilon$-net of $\mathcal{S}_p$.

**Lemma 8** (Lemma 1 in [17]). *For any bounded differentiable function $\Phi$ with bounded Lipschitz derivative, and any $\epsilon > 0$ we have:*

$$\lim_{n,p \to \infty} \left| \mathbb{E} \left[ \Phi \left( f_{\epsilon,\beta}(\boldsymbol{X}) \right) \right] - \mathbb{E} \left[ \Phi \left( f_{\epsilon,\beta}(\boldsymbol{G}) \right) \right] \right| = 0.$$

Subsequently, using classical arguments from both the theory of $\epsilon$-nets and statistical physics, the authors show that

$$\left| f_{\epsilon,\beta}(\boldsymbol{X}) - \widehat{\mathcal{R}}_n(\boldsymbol{\Theta}; \boldsymbol{X}, \boldsymbol{y}(\boldsymbol{X})) \right| \leq C_1(\epsilon) + \frac{C_2(\epsilon)}{\beta}, \qquad (33)$$

and the same inequality holds for $\boldsymbol{G}$. Since $C_1, C_2$ do not depend on $n, p$, it is possible to choose first $\epsilon$, then $\beta$ so that the RHS of (33) is as small as desired.

[17] therefore used the universality of the free energy as an intermediate step wards proving the universality of the training error. We generalize this result to the free energy defined for a general class of Boltzmann distributions, allowing us to prove universality in applications related to sampling.

## A.3 Sketch of proof of Lemma 8, adapted from [17]

**Interpolation path** For any $0 \leq t \leq \pi/2$, define

$$\boldsymbol{U}_t = \cos(t)\boldsymbol{X} + \sin(t)\boldsymbol{G}$$

Then $\boldsymbol{U}_t$ is a smooth interpolation path with independent columns, ranging from $\boldsymbol{U}_0 = \boldsymbol{X}$ to $\boldsymbol{U}_{\pi/2} = \boldsymbol{G}$. We can write, for any differentiable function $\psi$,

$$|\mathbb{E}\left[\psi(f_{\epsilon,\beta}(\boldsymbol{X}))\right] - \mathbb{E}\left[\psi(f_{\epsilon,\beta}(\boldsymbol{G}))\right]| \leq \int_0^{\pi/2} \left| \mathbb{E} \left[ \frac{d\psi(f_{\epsilon,\beta}(\boldsymbol{U}_t)))}{dt} \right] \right| dt,$$

and by the dominated convergence theorem it suffices to show that the integrand converges to $0$ for any $t$. The chain rule gives:

$$\frac{d\psi(f_{\epsilon,\beta}(\boldsymbol{U}_t)))}{dt} = \psi'(f_{\epsilon,\beta}(\boldsymbol{U}_t))) \left( \sum_{\mu=1}^{n} \left( \frac{d\boldsymbol{u}_{t,\mu}}{dt} \right)^\top \nabla_{\boldsymbol{u}_{t,\mu}} f_{\epsilon,\beta}(\boldsymbol{U}_t) \right), \tag{34}$$

and the dependency in $\psi$ can be easily controlled. Since all columns of $\boldsymbol{U}_t$ are i.i.d, we are left with showing

$$\lim_{n,p\to\infty} n\mathbb{E}_{(1)} \left[ \left( \frac{d\boldsymbol{u}_{t,1}}{dt} \right)^\top \nabla_{\boldsymbol{u}_{t,1}} f_{\epsilon,\beta}(\boldsymbol{U}_t) \right] = 0 \quad \text{a.s.}, \tag{35}$$

where $\mathbb{E}_{(1)}$ denotes the expectation with respect to $(\boldsymbol{x}_1, \boldsymbol{g}_1, \varepsilon_1)$.

**Showing** (35)   Imagine for a moment that $\boldsymbol{x}_1$ is Gaussian; then $\boldsymbol{u}_{t,1}$ and $d\boldsymbol{u}_{t,1}/dt$ are also jointly Gaussian, and we have

$$\mathbb{E} \left[ \left( \frac{d\boldsymbol{u}_{t,1}}{dt} \right)^\top \boldsymbol{u}_{t,1} \right] = \mathbb{E} \left[ (-\sin(t)\boldsymbol{x}_1 + \cos(t)\boldsymbol{g}_1)^\top (\cos(t)\boldsymbol{x}_1 + \sin(t)\boldsymbol{g}_1) \right]$$

$$= 0,$$

since $x_1$ and $g_1$ have the same covariance by definition. Therefore, they are independent, and we have

$$\mathbb{E}_{(1)} \left[ \left( \frac{d\boldsymbol{u}_{t,1}}{dt} \right)^\top \nabla_{\boldsymbol{u}_{t,1}} f_{\epsilon,\beta}(\boldsymbol{U}_t) \right] = \mathbb{E}_{(1)} \left[ \left( \frac{d\boldsymbol{u}_{t,1}}{dt} \right) \right]^\top \mathbb{E}_{(1)} \left[ \nabla_{\boldsymbol{u}_{t,1}} f_{\epsilon,\beta}(\boldsymbol{U}_t) \right] = 0.$$

On the other hand, it is possible to show that $\boldsymbol{x}_1$ only appears in (35) through scalar products with $\boldsymbol{\Theta}$ or $\boldsymbol{\Theta}^*$. As a result, we can leverage Assumption 4 to replace $\boldsymbol{x}_1$ by a Gaussian vector $\boldsymbol{w}$ independent from $\boldsymbol{g}_1$ as $p \to \infty$. Then, the reasoning above can be repeated with $\boldsymbol{w}$ and $\boldsymbol{g}_1$ to conclude the proof.

### A.4   Proof of Theorem 2

In order to prove our theorem 2, we now aim to adapt the proof from [17] to the following case where the distribution of $x$ can be a *mixture* of several other distributions, each with different mean and covariance. For a discrete set $\mathcal{C}$, we consider a family of distributions $(\nu_c)_{c\in\mathcal{C}}$ on $\mathbb{R}^p$, with means and covariances

$$\boldsymbol{\mu}_c = \mathbb{E}_{\mathbf{z}\sim\nu_c}[\mathbf{z}] \qquad \text{and} \qquad \boldsymbol{\Sigma}_c = \mathbb{E}_{\mathbf{z}\sim\nu_c}[\mathbf{z}\mathbf{z}^\top]$$

Given a type assignment $\sigma : [n] \to \mathcal{C}$, each sample $x_i$ is then drawn independently from $\nu_{\sigma(i)}$. The equivalent Gaussian model is straightforward: we simply take

$$\boldsymbol{g}_i \sim \mathcal{N}(\boldsymbol{\mu}_{\sigma(i)}, \boldsymbol{\Sigma}_{\sigma(i)}),$$

independently from each other. An important special case of this setting is when $\sigma$ is itself random, independently from the $\boldsymbol{x}_i$ and $\boldsymbol{g}_i$: the law of $\boldsymbol{g}_i$ is then a so-called Gaussian Mixture Model. Note that in the Gaussian mixture setting, the existence of the labeling function $\sigma$ implies that we coupled the labels for $\boldsymbol{X}$ and $\boldsymbol{G}$.

The main differences between our Assumptions 1-4 and Assumptions A1-A3 are the following:

 (i) Assumption 1 is unchanged.
 (ii) We relax (31) in Assumption 3 into

$$y_i = \eta_{\sigma(i)}(\boldsymbol{\Theta}^* \boldsymbol{x}_i, \varepsilon_i, c_i),$$

    when $\eta$ a Lipschitz function in its first two parameters. This allows in particular to incorporate classification problems in our setting, at no cost in the proof complexity.
 (iii) We assume a more general setup where the constraint set $\mathcal{S}_p$ is not necessarily a product set. This slight generalization will be useful while proving a reduction to multiple objectives in Theorem 4.
 (iv) We suppose that Assumptions 2 and 4 hold for any possible distribution $\nu_c$ for $c \in \mathcal{C}$ and its associated Gaussian equivalent model.
 (v) We allow the reference measures to be any sequence of Borel measures with support on $\mathcal{S}_p$, instead of only the Dirac measure on the $\epsilon$-net $\mathcal{N}_\epsilon$.

We now go through the proof of the previous section, highlighting the important changes.

**Free energy approximation**  This section goes basically unchanged; the approximation between $\widehat{\mathcal{R}}_n^*(\boldsymbol{X}, \boldsymbol{y}(\boldsymbol{X}))$ and $f_{\epsilon,\beta}(\boldsymbol{X})$ relies on Lipschitz arguments and concentration bounds on the $\boldsymbol{x}_i$ and $\boldsymbol{g}_i$, which are satisfied by our modification of Assumption 2.

**General Reference Measures**  Our proof for the universality of free energy in Theorem 2 is a generalization of the proof of Lemma 1 in [17]. Compactness of the supports ensures that the corresponding free energy:

$$f_{\beta,n}(\boldsymbol{Z}) = \int \exp\left(-\beta n R_n(\boldsymbol{\Theta}; \boldsymbol{Z}, \boldsymbol{y})\right) d\mu(\boldsymbol{\Theta}), \tag{36}$$

is finite for any $\boldsymbol{Z}$.

The corresponding Boltzmann measures can then be defined by setting the Radon-Nikodym derivative/density to be:

$$\frac{d\tilde{\mu}}{d\mu} = \exp\left(-\beta n \widehat{\mathcal{R}}_n(\boldsymbol{\Theta}; \boldsymbol{X}, \boldsymbol{y})\right). \tag{37}$$

The measure $\tilde{\mu}$ is then a Borel measure with support lying in $\mathcal{S}_p$. Therefore, through dominated convergence theorem, we can interchange differentiation and expectations w.r.t $\mu$ in the proof of Lemma 1 in [17]. For instance, equation (34) can be expressed as:

$$\mathbb{E}\left[\frac{\partial}{\partial t}\psi(f_{\beta,n}(\boldsymbol{U}_t))\right] = \mathbb{E}\left[\frac{\psi'(f_{\beta,n}(\boldsymbol{U}_t))}{n}\sum_{i=1}^{n}\frac{\int \widetilde{\boldsymbol{u}}_{t,i}^\top \widehat{\boldsymbol{d}}_{t,i}(\boldsymbol{\Theta})e^{-n\beta R_n(\boldsymbol{\Theta};\boldsymbol{Z},\boldsymbol{y})}d\mu_p}{\int e^{-n\beta R_n(\boldsymbol{\Theta};\boldsymbol{Z},\boldsymbol{y})}d\mu_p}\right]. \tag{38}$$

Similarly we substitute $\sum_{\boldsymbol{\Theta}}$ by $\int d\mu$ in the remaining arguments in the proof of Lemma 1 in [17].

**Interpolation path**  Recall that the important property of $\boldsymbol{U}_t$ is that

$$\mathbb{E}\left[\left(\frac{d\boldsymbol{U}_t}{dt}\right)^\top \boldsymbol{U}_t\right] = 0. \tag{39}$$

To this end, we set

$$\boldsymbol{u}_{t,i} = \boldsymbol{\mu}_{\sigma(i)} + \cos(t)(\boldsymbol{x}_i - \boldsymbol{\mu}_{\sigma(i)}) + \sin(t)(\boldsymbol{g}_i - \boldsymbol{\mu}_{\sigma(i)}),$$

and it is easy to check that (39) is satisfied. Another problem is that the columns of $\boldsymbol{U}_t$ are not i.i.d anymore, so we have to control

$$\frac{1}{n}\sum_{i=1}^{n}\left|\mathbb{E}_{(i)}\left[\left(\frac{d\boldsymbol{u}_{t,i}}{dt}\right)^\top \nabla_{\boldsymbol{u}_{t,i}}f_{\epsilon,\beta}(\boldsymbol{U}_t)\right]\right|, \tag{40}$$

where this time $\mathbb{E}_{(i)}$ is the expectation w.r.t $(\boldsymbol{x}_i, \boldsymbol{g}_i, \varepsilon_i)$. However, (40) is a weighted average over all values of $\sigma(\mu)$, and since $\mathcal{C}$ is finite is suffices to show (35) for any value of $\sigma(1)$.

**Showing (35)**  This section again relies on concentration properties of the $\boldsymbol{x}_i$ and $\boldsymbol{g}_i$, as well as Assumption 4. The arguments thus translate directly from [17].

## A.5  Proof of Proposition 3

We define the following free energy of the system:

$$f_{n,s,\epsilon}(\boldsymbol{X}, \boldsymbol{y}) = -\frac{1}{n}\log\int e^{-n\sum_{m=1}^{M}\beta_m\,\widehat{\mathcal{R}}_n^{(m)}(\boldsymbol{\Theta}^{(m)};\boldsymbol{X}^{(m)},\boldsymbol{y}^{(m)})-sn\,h(\boldsymbol{\Theta}^{(1)},...,\boldsymbol{\Theta}^{(M)})}d\mu_\epsilon^{1:M_1}d\mu^{(M_1+1):M}, \tag{41}$$

where the $\mu_m$ are the reference measures for the Boltzmann distributions in (16), and $\mu_\epsilon^m$ is the uniform measure supported on a minimal $\epsilon$-net of $\mathcal{S}_p^{(i)}$:

$$\mu_\epsilon^m = \frac{1}{|\mathcal{N}_\epsilon^m|}\sum_{\boldsymbol{\Theta}\in\mathcal{N}_\epsilon^m}\delta_{\boldsymbol{\Theta}}, \tag{42}$$

where $\delta_{\boldsymbol{\Theta}}$ denotes the Dirac measure at $\boldsymbol{\Theta}$. We establish the following result:

**Lemma 9** (Universality of the joint free energy). *Under Assumptions 1-4, for any fixed $\epsilon > 0$ and any bounded differentiable function $\psi$ with bounded Lipschitz derivative we have.*

$$\lim_{n \to \infty} |\mathbb{E}[\psi(f_{n,s,\epsilon}(\boldsymbol{X}, \boldsymbol{y}))] - \mathbb{E}[\psi(f_{n,s,\epsilon}(\boldsymbol{X}, \boldsymbol{y}))]| = 0.$$

*Proof.* We construct a reduction from the free energy of the form in equation 41 to the universality of free energy for a single objective in Theorem 1.

We construct an equivalent objective on the $pM \times kM$ dimensional space. Consider the following mapping:

$$\boldsymbol{\Theta} = \begin{pmatrix} \boldsymbol{\Theta}^{(1)} & \boldsymbol{0} & \boldsymbol{0} & \cdots \\ \boldsymbol{0} & \boldsymbol{\Theta}^{(2)} & \boldsymbol{0} & \cdots \\ \cdots & \cdots & \cdots & \cdots \\ \boldsymbol{0} & \boldsymbol{0} & \cdots & \boldsymbol{\Theta}^{(M)} \end{pmatrix} \tag{43}$$

. Let $\mathcal{S}_{pM}$ be the set obtained by applying the mapping in equation 43 to $\mathcal{S}_p^{(1)}, \cdots, \mathcal{S}_p^{(M)}$. Let $\boldsymbol{X} = (\boldsymbol{X}^{(1)}, \cdots, \boldsymbol{X}^{(M)})$ denote the combined input matrix with each row of dimension $pM$. We note that $\mathcal{S}_{pM}$ is a product of $kM$ compact sets, each satisfying the assumption 2. Similarly, we have the combined output vector:

$$\boldsymbol{y} = \begin{pmatrix} \boldsymbol{y}^{(1)} \\ \vdots \\ \boldsymbol{y}^{(M)} \end{pmatrix} \tag{44}$$

We define $\ell : \mathbb{R}^{k'M} \times \mathbb{R}^{kM} \to \mathbb{R}$ by:

$$\ell(\boldsymbol{u}, \mathbf{y}) = \sum_{m=1}^{M} \beta_m \ell_m(\boldsymbol{u}[(k-1)m : km], \mathbf{y}[(k-1)m : km]). \tag{45}$$

Similarly, we define the total regularization as:

$$r(\boldsymbol{\Theta}) = \sum_{m=1}^{M} \beta_m r_m(\boldsymbol{\Theta}[(m-1)p : mp, (m-1) : k, mk]) \tag{46}$$
$$+ sh(\boldsymbol{\Theta}[0 : p, 0 : k], \cdots, \boldsymbol{\Theta}[(M-1)p : Mp, (M-1)k : Mk]).$$

Let $\widehat{\mathcal{R}}_n(\boldsymbol{\Theta}; \boldsymbol{X}, \boldsymbol{y})$ denote the following objective on the combined vector $\boldsymbol{\Theta}$:

$$\widehat{\mathcal{R}}_n(\boldsymbol{\Theta}; \boldsymbol{X}, \boldsymbol{y}) = \frac{1}{n} \sum_{i=1}^{n} \ell(\boldsymbol{\Theta}^\top \boldsymbol{x}_i, \mathbf{y}_i) + r(\boldsymbol{\Theta}) \tag{47}$$

Then, using all the definitions above, we have

$$\widehat{\mathcal{R}}_n(\boldsymbol{\Theta}; \boldsymbol{Z}, \boldsymbol{y}) = \sum_{m=1}^{M} \beta_m \widehat{\mathcal{R}}_n^{(m)}(\boldsymbol{\Theta}^{(m)}; \boldsymbol{X}^{(m)}, \boldsymbol{y}^{(m)}) + s\, h(\boldsymbol{\Theta}^{(1)}, \cdots, \boldsymbol{\Theta}^{(M)}) \tag{48}$$

Therefore, we obtain that:

$$f_{n,s,\epsilon}(\boldsymbol{X}, \boldsymbol{y}) = -\frac{1}{n} \log \int e^{-n \sum_{m=1}^{M} \beta_m \widehat{\mathcal{R}}_n^{(m)}(\boldsymbol{\Theta}^{(m)}; \boldsymbol{X}^{(m)}, \boldsymbol{y}^{(m)}) - sn\, h(\boldsymbol{\Theta}^{(1)}, \ldots, \boldsymbol{\Theta}^{(M)})} d\mu_\epsilon^{1:M_1} d\mu^{(M_1+1):M}$$

$$= -\frac{1}{n} \log \int e^{-n\widehat{\mathcal{R}}_n(\boldsymbol{\Theta}; \boldsymbol{X}, \boldsymbol{y})} d\mu_\epsilon^{1:M_1} d\mu^{(M_1+1):M}. \tag{49}$$

We further note that the constraint sets on $\boldsymbol{\Theta}$, and the joint means, covariances on $\boldsymbol{X}$ satisfy the assumptions A2. Therefore, using Theorem 1, we obtain that:

$$\lim_{n \to \infty} |\mathbb{E}[\psi(f_{n,s,\epsilon}(\boldsymbol{X}, \boldsymbol{y}))] - \mathbb{E}[\psi(f_{n,s,\epsilon}(\boldsymbol{X}, \boldsymbol{y}))]| = 0$$

$\square$

The proof of Proposition 3 then follows using the $\epsilon$-net approximation in (33) for $\boldsymbol{\Theta}[1 : M_1]$.

### A.6 Proof of Theorem 4

Our proof relies on the following result:

**Lemma 10.** *For any $n \in \mathbb{N}$, $q_n(s)$ is a concave function of $s$, that is differentiable at $0$, and*

$$q'_n(0) = \mathbb{E}\left[\left\langle h\left(\boldsymbol{\Theta}^{(1)}, \cdots, \boldsymbol{\Theta}^{(M)}\right)\right\rangle_{\boldsymbol{G}}\right]. \tag{50}$$

*Proof.* Consider the joint free energy in Equation (18):

$$f_{n,s}(\boldsymbol{\Theta}[1:M_1], \boldsymbol{X}, \boldsymbol{y}) = -\frac{1}{n}\log\int e^{-sn\ h(\boldsymbol{\Theta}^{(1)},\ldots,\boldsymbol{\Theta}^{(M)})}dP^{(M_1+1):M}, \tag{51}$$

Let $\langle\cdot\rangle_{M_1+1:M}$ denote the expectation w.r.t the product measure

$$d\tilde{\mu}(\boldsymbol{\Theta}[1:M_1], \boldsymbol{X}, \boldsymbol{y}) = e^{-sn\ h(\boldsymbol{\Theta}^{(1)},\ldots,\boldsymbol{\Theta}^{(M)})}dP^{(M_1+1):M}.$$

We observe that:

$$\frac{df_{n,s}}{ds}(\boldsymbol{\Theta}[1:M_1], \boldsymbol{X}, \boldsymbol{y}) = \langle h(\boldsymbol{\Theta}^{(1)}, \ldots, \boldsymbol{\Theta}^{(M)})\rangle_{M_1+1:M}. \tag{52}$$

Differentiating w.r.t s again, and using the dominated convergence theorem, we obtain:

$$-\frac{1}{n}\frac{d^2 f_{n,s}}{ds^2}(\boldsymbol{\Theta}[1:M_1], \boldsymbol{X}, \boldsymbol{y}) = \langle h(\boldsymbol{\Theta}^{(1)}, \ldots, \boldsymbol{\Theta}^{(M)})^2\rangle_{M_1+1:M} - \langle h(\boldsymbol{\Theta}^{(1)}, \ldots, \boldsymbol{\Theta}^{(M)})\rangle^2_{M_1+1:M}. \tag{53}$$

Since the R.H.S equals the variance of the variable $h(\boldsymbol{\Theta}^{(1)}, \ldots, \boldsymbol{\Theta}^{(M)})$ w.r.t $\tilde{\mu}^{(M_1+1):M}$, we have:

$$\frac{d^2 f_{n,s}(\boldsymbol{\Theta}[1:M_1], \boldsymbol{Z}, s)}{ds^2} \leq 0. \tag{54}$$

Therefore, for fixed $\boldsymbol{\Theta}[1:M_1]$, we obtain that the function:

$$\widehat{\mathcal{R}}_{n,s}(\boldsymbol{\Theta}[1:M_1], \boldsymbol{X}, \boldsymbol{y}) = \sum_{m=1}^{M_1} \widehat{\mathcal{R}}_n^{(m)}(\boldsymbol{\Theta}^{(m)}; \boldsymbol{X}^{(m)}, \boldsymbol{y}^{(m)}) + f_{n,s}(\boldsymbol{\Theta}[1:M_1], \boldsymbol{X}, \boldsymbol{y}). \tag{55}$$

is concave in s. Next, we recall that pointwise infimum of arbitrary collections of concave functions is concave [48]. Therefore, we obtain that the function:

$$q_n(s) = \mathbb{E}\left[\min_{\boldsymbol{\Theta}[1:M_1]} \widehat{\mathcal{R}}_{n,s}(\boldsymbol{\Theta}[1:M_1], \boldsymbol{G}, \boldsymbol{y})\right], \tag{56}$$

is concave in $s$. Then, by Danskin's theorem, the subdifferential of $q_n$ at zero is the set

$$\left\{\langle h(\boldsymbol{\Theta}^{(1)}, \ldots, \boldsymbol{\Theta}^{(M)})^2\rangle_{M_1+1:M}, \boldsymbol{\Theta}[1:M_1] \in \arg\min \widehat{\mathcal{R}}_{n,0}(\boldsymbol{\Theta}[1:M_1], \boldsymbol{X}, \boldsymbol{y})\right\}$$

But by Assumption 6, this set only has one element, and hence $q_n$ is differentiable at 0. $\square$

Next, we relate the convergence of the above functions to the expectation of $h(\boldsymbol{\Theta}^{(1)}, \ldots, \boldsymbol{\Theta}^{(M)})$, through the following standard result from Convex Analysis:

**Theorem 11.** *(Theorem 25.7. in [48]): Let $C$ be an open convex set, and let $f$ be a convex function which is finite and differentiable on $C$. Let $f_1, f_2, \ldots$, be a sequence of convex functions finite and differentiable on $C$ such that $\lim_{n\to\infty} f_n(x) = f(x)$ for every $x \in C$. Then*

$$\lim_{n\to\infty} \nabla f_n(x) = \nabla f(x), \quad \forall x \in C.$$

By Assumption 5,

$$\lim_{n\to\infty} q_{g,n}(s) = q(s). \tag{57}$$

Applying theorem 11 to the sequence $q_{g,n}(s)$ yields:

$$\lim_{n\to\infty} q'_n(0) = q'(0). \tag{58}$$

Now, consider the corresponding free energy for the data distribution $p_{\boldsymbol{x}}$:

$$q_{x,n}(s) = \mathbb{E}\left[\min_{\boldsymbol{\Theta}[1:M_1]} \widehat{\mathcal{R}}_{n,s}(\boldsymbol{\Theta}[1:M_1], \boldsymbol{X}, \boldsymbol{y})\right]. \tag{59}$$

We again have that $q_{x,n}(s)$ is a concave, differentiable function in $s$, with:

$$q'_{x,n}(0) = \mathbb{E}\left[\left\langle h\left(\boldsymbol{\Theta}^{(1)}, \cdots, \boldsymbol{\Theta}^{(M)}\right)\right\rangle_{\boldsymbol{X}}\right]. \tag{60}$$

Now, Proposition 3 and equation (57) imply that

$$\lim_{n\to\infty} q_{x,n}(s) = \lim_{n\to\infty} q_n(s) = q(s). \tag{61}$$

Therefore, Theorem 11 applied to the sequence of functions $q_{x,n}(s)$ implies that:

$$\lim_{n\to\infty} q'_{x,n}(0) = q'(0). \tag{62}$$

By equations 60 and Lemma 10, we then obtain:

$$\lim_{n,p\to\infty}\left|\mathbb{E}\left[\left\langle h\left(\boldsymbol{\Theta}^{(1)}, \cdots, \boldsymbol{\Theta}^{(M)}\right)\right\rangle_{\boldsymbol{X}}\right] - \mathbb{E}\left[\left\langle h\left(\boldsymbol{\Theta}^{(1)}, \cdots, \boldsymbol{\Theta}^{(M)}\right)\right\rangle_{\boldsymbol{G}}\right]\right| = 0. \tag{63}$$

## A.7 Proof of Theorem 6: One-dimensional CLT for Random Feature Models

Our proof relies on a reduction to Theorem 2 in [16] and the proof of corollary 2 in [17]. We first notice that it suffices to show the result when $\boldsymbol{z} \sim \mathcal{N}(\boldsymbol{\mu}^{\boldsymbol{z}}, \boldsymbol{\Sigma}^{\boldsymbol{z}})$ is a Gaussian variable; and upon rescaling of $\sigma$ we shall assume that $\mathrm{tr}(\boldsymbol{\Sigma}^{\boldsymbol{z}}) = p$.

Let $\boldsymbol{V} = \boldsymbol{\Sigma}^{\boldsymbol{z}1/2}\boldsymbol{F}$, and define the following events:

$$\mathcal{A}_1 = \left\{\sup_{i,j\in[d]} \left|\boldsymbol{v}_i^\top \boldsymbol{v}_j - \delta_{ij}\right| \le C_1 \left(\frac{\log d}{d}\right)^{1/2}\right\} \qquad \mathcal{A}_2 = \left\{\sum_{i\in[d]} \left|\|\boldsymbol{v}_i\|^2 - 1\right| \le C_2\right\}$$

$$\mathcal{A}_3 = \{\|\boldsymbol{F}\|_{\mathrm{op}} \le C_3\} \qquad\qquad\qquad \mathcal{A}_4 = \{\|\boldsymbol{V}\|_{\mathrm{op}} \le C_4\}$$

Since the $\boldsymbol{f}_i$ are independent and sub-gaussian, Lemma 22 in [17] implies that there exists constants $C_1, C_2, C_3, C_4$ such that $\mathcal{B} = \mathcal{A}_1 \cap \mathcal{A}_2 \cap \mathcal{A}_3 \cap \mathcal{A}_4$ is a high-probability event. Now, for $i \in [d]$, we define

$$\sigma_i(u) = \sigma(u + \boldsymbol{f}_i^\top \boldsymbol{\mu}^{\boldsymbol{z}}). \tag{64}$$

Now, as in [17], we argue that the proof of Theorem 2 in [16] still applies to our setting. Indeed:

- since $\boldsymbol{z}$ does not have identity covariance, we replace the conditions on $\boldsymbol{F}$ by the exact same ones on $\boldsymbol{V}$,
- the Stein method they use proceeds term by term, so using a different $\sigma_i$ in each term does not matter as long as they satisfy the boundedness assumptions above,
- since we match the means of $\boldsymbol{g}$ and those of $\boldsymbol{x}$, the requirement that $\sigma$ be odd is unimportant in our setting.

In particular, for bounded Lipschitz test functions $\varphi$, the proof of Lemma 2 in [16] shows that for any $\boldsymbol{\theta} \in \mathbb{R}^d$,

$$\left|\mathbb{E}\left[\varphi(\boldsymbol{\theta}^\top \boldsymbol{x})\right] - \mathbb{E}\left[\varphi(\boldsymbol{\theta}^\top \boldsymbol{g})\right]\right| \le \frac{C\|\boldsymbol{\theta}\|_\infty \operatorname{polylog}(p)}{\nu^2} \tag{65}$$

where $\nu^2$ is the variance of $\boldsymbol{\theta}^\top \boldsymbol{x}$:

$$\nu^2 = \boldsymbol{\theta}^\top \boldsymbol{\Sigma}\boldsymbol{\theta}. \tag{66}$$

We now place ourselves in the setting where $\boldsymbol{\theta} \in \mathcal{S}_p$ where $\mathcal{S}_p$ is defined in (26), and we consider two cases:

(i) if $\nu^2 > p^{-2\eta/3}$, then (65) reduces to

$$\left|\mathbb{E}\left[\varphi(\boldsymbol{\theta}^\top \boldsymbol{x})\right] - \mathbb{E}\left[\varphi(\boldsymbol{\theta}^\top \boldsymbol{g})\right]\right| \le \frac{C \operatorname{polylog}(p)}{p^{\eta/3}}. \tag{67}$$

(ii) if instead $\nu^2 > p^{-2\eta/3}$, then by the Lipschitz property of $\varphi$ we have

$$\left| \mathbb{E}\left[\varphi(\boldsymbol{\theta}^\top \boldsymbol{x})\right] - \varphi(\boldsymbol{\mu}) \right| \leq C\sqrt{\nu^2} = \frac{C}{p^{\eta/3}}, \tag{68}$$

and the same holds for $\boldsymbol{g}$.

In both cases, the bounds goes to 0 uniformly over the whole constraint set $\mathcal{S}_p$, which shows that Assumption 4 holds.

We now move to checking Assumption 7; Lemma 8 in [16] (more precisely, eq. (159)) exactly shows that for $\boldsymbol{\theta} \in \mathcal{S}_p$, the random variable $\boldsymbol{\theta}^\top x - \boldsymbol{\theta}^\top \boldsymbol{\mu}$ is $C$-subgaussian for an absolute constant $C$. Hence, we only need to show that $\boldsymbol{\mu}$ is uniformly bounded. Recall that

$$\mu_i = \mathbb{E}\left[\sigma(\boldsymbol{f}_i^\top \boldsymbol{z})\right] = \mathbb{E}\left[\sigma\left(\boldsymbol{f}_i^\top \boldsymbol{\mu}^z + (1 + \tau_i)\tilde{z}\right)\right]$$

where $\tilde{z}$ is a standard normal variable and

$$\tau_i = \|\boldsymbol{v}_i\| - 1.$$

Then, we can write using the Lipschitz property of $\sigma$

$$\sigma(\boldsymbol{f}_i^\top \boldsymbol{z}) = \sigma(\tilde{z}) + (\boldsymbol{f}_i^\top \boldsymbol{\mu}^z + \tau_i \tilde{z})\tilde{\sigma}(\boldsymbol{f}_i^\top \boldsymbol{z})$$

where $\tilde{\sigma}$ is a uniformly bounded function. By assumption, $\sigma(\tilde{z})$ has zero mean, and hence by the Cauchy-Schwarz inequality

$$\|\boldsymbol{\mu}\|^2 \leq \sum_{i \in [d]} (\boldsymbol{f}_i^\top \boldsymbol{\mu}^z)^2 + (\|\boldsymbol{v}_i\| - 1)^2$$

$$\leq \|\boldsymbol{F}\|_{\text{op}}^2 \|\boldsymbol{\mu}^z\|^2 + \sum_{i \in [d]} \left| \|\boldsymbol{v}_i\|^2 - 1 \right|$$

$$\leq C$$

under the high-probability event $\mathcal{B}$. $\qquad \square$

## A.8  Proof of Theorem 5

Our proof utilizes the results in [12] that describe the asymptotic limits of the estimators obtained through empirical risk minimization on the mixture of gaussians dataset. We note that the assumptions A1-A5 of their Theorem 1 are satisfied by our setting.

Let $\boldsymbol{W}^\star$ denote the minimizer of the objective in equation 22, and let $\boldsymbol{Z}^\star = \boldsymbol{X}\boldsymbol{W}^\star$. Let $\boldsymbol{\xi}_{k\in[K]} \sim \mathcal{N}(\boldsymbol{0}, \boldsymbol{I}_K)$, $\boldsymbol{\Xi}_k \in \mathbb{R}^{K \times d}$ be sets of $K$-dimensional vectors and dimensional matrices respectively, with i.i.d entries sampled from $\mathcal{N}(0, 1)$.

Then, Theorem 1 in [12] proves that for any pseudo-lipschitz functions of finite order, $\phi_1 : \mathbb{R}^{K \times d} \to \mathbb{R}, \phi_2 : \mathbb{R}^{K \times n} \to \mathbb{R}$:

$$\phi_1(\boldsymbol{W}^\star) \xrightarrow[n,d\to+\infty]{P} \mathbb{E}_{\boldsymbol{\Xi}}\left[\phi_1(\boldsymbol{G})\right], \qquad \phi_2(\boldsymbol{Z}^\star) \xrightarrow[n,d\to+\infty]{P} \mathbb{E}_{\boldsymbol{\xi}}\left[\phi_2(\boldsymbol{H})\right] \tag{69}$$

Here $\boldsymbol{G}$ and $\boldsymbol{H}$ are functions of certain finite dimensional parameters

$$\boldsymbol{u} := (\boldsymbol{Q}_k \in \mathbb{R}^{K \times K}, \boldsymbol{M}_k \in \mathbb{R}^K, \boldsymbol{V}_k \in \mathbb{R}^{K \times K}, \hat{\boldsymbol{Q}}_k \in \mathbb{R}^{K \times K}, \hat{\boldsymbol{m}}_k \in \mathbb{R}^K, \hat{\boldsymbol{V}}_k \in \mathbb{R}^{K \times K})_{k \in [K]}$$

and the random vectors $\boldsymbol{\xi}_{k\in[K]}, \boldsymbol{\Xi}_{k\in[K]}$. The matrix $\boldsymbol{H}$ is obtained by concatenating the following functions $\boldsymbol{h}_k$, $\rho_k n$ time for each $k$:

$$\boldsymbol{h}_k = \boldsymbol{V}_k^{1/2} \text{Prox}_{\ell(\boldsymbol{e}_k, \boldsymbol{V}_k^{1/2}\bullet)}(\boldsymbol{V}_k^{-1/2}\boldsymbol{\omega}_k) \in \mathbb{R}^K, \qquad \boldsymbol{\omega}_k \equiv \boldsymbol{M}_k + \boldsymbol{b} + \boldsymbol{Q}_k^{1/2}\boldsymbol{\xi}_k, \tag{70}$$

Similarly, the matrix $\boldsymbol{G} \in \mathbb{R}^{K \times d}$ is described by:

$$\boldsymbol{G} = \mathbf{A}^{\frac{1}{2}} \odot \text{Prox}_{r(\mathbf{A}^{\frac{1}{2}}\odot\bullet)}(\mathbf{A}^{\frac{1}{2}} \odot \boldsymbol{B}), \quad \mathbf{A}^{-1} \equiv \sum_k \hat{\boldsymbol{V}}_k \otimes \boldsymbol{\Sigma}_k, \quad \boldsymbol{B} \equiv \sum_k \left( \boldsymbol{\mu}_k \hat{\boldsymbol{m}}_k^\top + \boldsymbol{\Xi}_k \odot \sqrt{\hat{\boldsymbol{Q}}_k \otimes \boldsymbol{\Sigma}_k} \right).$$

Further define the function: $\boldsymbol{f}_k \equiv \boldsymbol{V}_k^{-1}(\boldsymbol{h}_k - \boldsymbol{\omega}_k)$. The equivalent bias vector $\boldsymbol{b}^\star$ is defined through the linear equation:

$$\sum_k \rho_k \mathbb{E}_{\boldsymbol{\xi}}\left[\boldsymbol{V}_k \boldsymbol{f}_k\right] = \boldsymbol{0}, \tag{71}$$

and is therefore, unique, differentiable in $\boldsymbol{u}$. The parameters $\boldsymbol{u}$ satisfy the following equations:

$$\begin{cases} \boldsymbol{Q}_k = \frac{1}{d}\mathbb{E}_{\boldsymbol{\Xi}}[\boldsymbol{G}\boldsymbol{\Sigma}_k\boldsymbol{G}^\top] \\ \boldsymbol{M}_k = \frac{1}{\sqrt{d}}\mathbb{E}_{\boldsymbol{\Xi}}[\boldsymbol{G}\boldsymbol{\mu}_k] \\ \boldsymbol{V}_k = \frac{1}{d}\mathbb{E}_{\boldsymbol{\Xi}}\left[\left(\boldsymbol{G}\odot\left(\hat{\boldsymbol{Q}}_k\otimes\boldsymbol{\Sigma}_k\right)^{-\frac{1}{2}}\odot\left(\boldsymbol{I}_K\otimes\boldsymbol{\Sigma}_k\right)\right)\boldsymbol{\Xi}_k^\top\right] \end{cases} \qquad \begin{cases} \hat{\boldsymbol{Q}}_k = \alpha\rho_k\mathbb{E}_{\boldsymbol{\xi}}\left[\boldsymbol{f}_k\boldsymbol{f}_k^\top\right] \\ \hat{\boldsymbol{V}}_k = -\alpha\rho_k\boldsymbol{Q}_k^{-\frac{1}{2}}\mathbb{E}_{\boldsymbol{\xi}}\left[\boldsymbol{f}_k\boldsymbol{\xi}^\top\right] \\ \hat{\boldsymbol{m}}_k = \alpha\rho_k\mathbb{E}_{\boldsymbol{\xi}}\left[\boldsymbol{f}_k\right]. \end{cases} \tag{72}$$

.

We observe that the system of equations (72) can be expressed as a multi-dimensional fixed point equation:

$$\boldsymbol{u} = F_n(\boldsymbol{u}). \tag{73}$$

We make the following assumption, which is slightly stronger than (A5) in [12]:

**Assumption A5.** *The fixed point equations $\boldsymbol{u} = F_n(\boldsymbol{u})$ have unique solutions $\forall n \in \mathbb{N}$. Let $\hat{\boldsymbol{u}}_n$ be the unique solution to $\boldsymbol{u} = F_n(\boldsymbol{u})$. We further assume the solutions are uniformly bounded, i.e:*

$$\|\hat{\boldsymbol{u}}_n\| \leq K \tag{74}$$

*for some constant $K$, and the jacobian of the fixed point equations $I - \frac{dF_n}{d\boldsymbol{u}}$ is invertible. Furthermore, we assume that the same assumptions hold for the limiting equations $\boldsymbol{u} = F(\boldsymbol{u})$.*

**Remark**: While we assume the above conditions, as noted in [12], the fixed point equations 72, correspond to the optimality conditions of a strictly convex-concave problem. This can be rigorously proven using the properties of Bregman envelopes, as in [18, 49]. The strict convexity-concavity then implies the uniqueness of the fixed points as well as the differentiability of the limits.

We now prove the following result:

**Lemma 12.** *Under assumption 8, the system of equations (72) converge uniformly to a limiting system of equations $\boldsymbol{u} = F(\boldsymbol{u})$.*

*Proof.* We first show that the coordinates of the equivalent minimizer $\boldsymbol{G}$ can be expressed as follows:

$$\boldsymbol{G}_i = g(\{\mu_{c,i}\}_{c\in\mathcal{C}}, \{\sigma_{c,i}\}_{c\in\mathcal{C}}, \xi_i, \boldsymbol{u}) \tag{75}$$

Where $g$ is differentiable function and $\xi_i$ denote independent Gaussian random variables. Indeed, from the separability assumption on $r$, and the definition of the prox operator, we have

$$\text{Prox}_{r(\mathbf{A}^{\frac{1}{2}}\odot\bullet)}(\mathbf{A}^{\frac{1}{2}}\odot\boldsymbol{B}) = \underset{\mathbf{z}}{\arg\min}\, r(\mathbf{A}^{1/2}\odot\mathbf{z}) + \frac{1}{2}\|\mathbf{z} - (\mathbf{A}^{1/2}\odot\boldsymbol{B})\|^2 \tag{76}$$

$$= \underset{\mathbf{z}}{\arg\min}\sum_{i=1}^d \psi_r((\mathbf{A}^{1/2}\odot\mathbf{z})_i) + \frac{1}{2}\sum_{i=1}^d (\mathbf{z}_i - (\mathbf{A}^{1/2}\odot\boldsymbol{B})_i)^2. \tag{77}$$

$\boldsymbol{u}$ therefore only depends on the $i_{th}$ entry of $(\mathbf{A}^{1/2}\odot\boldsymbol{B})$. Further, since all $\boldsymbol{\Sigma}_c$ are assumed diagonal, the entries of $(\mathbf{A}^{1/2}\odot\mathbf{z})_i$ and $(\mathbf{A}^{1/2}\odot\boldsymbol{B})_i$ only depend on $\mathbf{z}_i, \{\mu_{c,i}\}_{c\in\mathcal{C}}, \{\sigma_{c,i}\}_{c\in\mathcal{C}}, \xi_i$, and the parameters $\boldsymbol{u}$. The differentiability of $g$ then follows from the Implicit function theorem, applied to $\psi_r(\mathbf{A}^{1/2}\odot\bullet) + 1/2(\bullet - (\mathbf{A}^{1/2}\odot\boldsymbol{B})_i)^2$. The same holds for the matrix $\boldsymbol{H}$.

We next observe that each coordinate of $F_n$ of the above system of equations 72 can be expressed as an expectation of a fixed continuous function w.r.t the joint empirical measure of $(\{\mu_{c,i}\}_{c\in\mathcal{C}}, \{\sigma_{c,i}\}_{c\in\mathcal{C}})$. For instance, consider the $(i,j)_{th}$ entry of $\boldsymbol{Q}_k$. We have:

$$\boldsymbol{Q}_{k,ij} = F_{q,i,j,n} = \frac{1}{d}\sum_{\ell=1}^d \mathbb{E}_{\boldsymbol{\Xi}}[\boldsymbol{G}_{i\ell}(\boldsymbol{\Sigma}_k)_{\ell\ell}\boldsymbol{G}_{\ell j}]. \tag{78}$$

Using equation (75), we have that $\boldsymbol{G}_{i\ell}$ only depends on the $\ell_{th}$ coordinates of the means, covariances. Therefore, for fixed $\hat{\boldsymbol{u}}_n$, $\boldsymbol{Q}_{k,ij}$ is an expectation w.r.t the joint empirical measure of $(\{\mu_{c,i}\}_{c\in\mathcal{C}}, \{\sigma_{c,i}\}_{c\in\mathcal{C}})$ of a continuous function. By Assumption 8, we have that

$$\lim_{n\to\infty} F_{q,i,j,n} = F_{q,i,j},$$

where $F_{q,i,j}$ denotes the expectation of $E_{\boldsymbol{\Xi}}[\boldsymbol{G}_{i\ell}(\boldsymbol{\Sigma}_k)_{\ell\ell}\boldsymbol{G}_{\ell j}]$ w.r.t the joint empirical measure.

To show that the convergence is uniform, we utilize assumptions 7 and A5. Since each term in $F_n$ can be expressed as:

$$F_n^j(\boldsymbol{u}) = \frac{1}{d}\sum_{i=1}^{d}\Phi_j(\boldsymbol{u}, \{\mu_{c,i}\}_{c\in\mathcal{C}}, \{\sigma_{c,i}\}_{c\in\mathcal{C}}), \tag{79}$$

for some function $\Phi_j$ with Lipschitz constant $L_j$. Therefore, for any $n \in \mathbb{N}$, $F_n^j(\boldsymbol{u})$ is $L_j$ Lipschitz. This implies the uniform convergence of $F_n$ to $F$. $\qquad\square$

We now use the above result to prove convergence of the sequence of solutions $\hat{\boldsymbol{u}}_n$.

**Lemma 13.** *Under Assumptions 8 and A5:*

$$\lim_{n\to\infty}\hat{\boldsymbol{u}}_n = \hat{\boldsymbol{u}}, \tag{80}$$

*where $\hat{\boldsymbol{u}}$ is the solution to the limiting equations $\boldsymbol{u} = F(\boldsymbol{u})$.*

*Proof.* By assumption, $\hat{\boldsymbol{u}}_n$ are bounded. Therefore, by the Bolzano–Weierstrass theorem, there exists a convergent subsequence. Let $\hat{\boldsymbol{u}}_{n_j}$ be any such subsequence with corresponding limit $\tilde{\boldsymbol{u}}$. We have:

$$\|F(\tilde{\boldsymbol{u}}) - \tilde{\boldsymbol{u}}\| \leq \|F(\tilde{\boldsymbol{u}}) - F(\hat{\boldsymbol{u}}_{n_j})\| + \|F(\hat{\boldsymbol{u}}_{n_j}) - F_{n_j}(\hat{\boldsymbol{u}}_{n_j})\|.$$

By uniform convergence of $F_n$ to $F$ (Lemma 12, we have that $\|F(\hat{\boldsymbol{u}}_{n_j}) - F_{n_j}(\hat{\boldsymbol{u}}_{n_j})\| \to 0$ while $\|F(\tilde{\boldsymbol{u}}) - F(\hat{\boldsymbol{u}}_{n_j})\| \to 0$ from the convergence of $\hat{\boldsymbol{u}}_{n_j}$ to $\tilde{\boldsymbol{u}}$ and the continuity of $F$. Therefore, we must have $F(\tilde{\boldsymbol{u}}) - \tilde{\boldsymbol{u}} = 0$ and thus $\tilde{\boldsymbol{u}} = \hat{\boldsymbol{u}}$. Therefore, any convergence subsequence of $\hat{\boldsymbol{u}}_n$ converges to $\hat{\boldsymbol{u}}$. Since $\hat{\boldsymbol{u}}_n$ are bounded, this implies that $\lim_{n\to\infty}\hat{\boldsymbol{u}}_n = \hat{\boldsymbol{u}}$. $\qquad\square$

Now, let $\Phi$ be a twice differentiable test function with a Hessian having bounded spectral norm. We define:

$$h_d(\boldsymbol{W}) = \frac{1}{d}\sum_{i=1}^{d}\Phi(\{W_{c,i}\}_{c\in\mathcal{C}}, \{\mu_{c,i}\}_{c\in\mathcal{C}}, \{\sigma_{c,i}\}_{c\in\mathcal{C}}). \tag{81}$$

Let $\hat{\boldsymbol{u}}(s)$ denote the solution to the limiting equation $\boldsymbol{u} = F_s(\boldsymbol{u})$ defined in Lemma 12 for regularization

$$r_s(\boldsymbol{W}) = \frac{\lambda}{n}r_d(\boldsymbol{W}) + sh_d(\boldsymbol{W})$$

By Assumption A5, $\boldsymbol{I} - \frac{dF_s}{d\boldsymbol{u}}$ is invertible in a neighbourhood of 0. Therefore, by implicit function theorem and uniqueness of the fixed points, $\hat{\boldsymbol{u}}(s)$, is a continuously differentiable function of $s$.

Using equation (75), we obtain that $G$ is a differentiable function of $s$. We now consider the perturbed training loss:

$$\widehat{\mathcal{R}}(\boldsymbol{W}, \boldsymbol{b}, s) = \frac{1}{n}\sum_{i=1}^{n}\ell\left(\frac{\boldsymbol{W}\boldsymbol{x}_i}{\sqrt{d}} + \boldsymbol{b}, \boldsymbol{y}_i\right) + r_s(\boldsymbol{W}). \tag{82}$$

By the boundedness of the Hessian of $\Phi$, $\widehat{\mathcal{R}}(\boldsymbol{W}, \boldsymbol{b}, s)$ satisfies the assumptions of strict convexity, coercivity for small enough $s$. We first note from (69) and Theorem 2 in [12], the expected training error converges to the following limit:

$$\mathbb{E}_{\boldsymbol{X}}\left[\frac{1}{n}\sum_{i=1}^{n}\ell\left(\frac{\boldsymbol{W}\boldsymbol{x}_i}{\sqrt{d}} + \boldsymbol{b}, \boldsymbol{y}_i\right)\right] \xrightarrow[n,d\to+\infty]{} \sum_{k=1}^{K}\rho_k\mathbb{E}_{\boldsymbol{\xi}}[\ell(\boldsymbol{e}_k, \boldsymbol{h}_k)]. \tag{83}$$

Similar to equation (75), we have that $\boldsymbol{h}_k$ differentiable functions of $\hat{\boldsymbol{u}}(s)$.

From Assumption 9 and the form of the perturbation 81, we further have that $\lambda r(\boldsymbol{W}) + sh(\boldsymbol{W})$ is a pseudo-Lipschitz function of $\boldsymbol{W}$ of finite order Therefore, Equation (69) gives:

$$\mathbb{E}_{\boldsymbol{X}}\left[\frac{1}{d}\lambda r(\boldsymbol{W}^*) + sh(\boldsymbol{W}^*)\right] \xrightarrow[n,d\to+\infty]{} \mathbb{E}_{\boldsymbol{\Xi}}\left[\frac{1}{d}\lambda r_d(\boldsymbol{G}_n) + sh_d(\boldsymbol{G}_n)\right]. \tag{84}$$

Define:

$$q_n(s, \boldsymbol{u}) = \frac{1}{d} \sum_{i=1}^{d} \mathbb{E}_{\boldsymbol{\Xi}} \left[ \psi_r(g(\{\mu_{c,i}\}_{c \in \mathcal{C}}, \{\sigma_{c,i}\}_{c \in \mathcal{C}}, \boldsymbol{\Sigma}_i^k, \xi_i, \boldsymbol{u}_n)) \right] \tag{85}$$

$$+ s \frac{1}{d} \sum_{i=1}^{d} \mathbb{E}_{\boldsymbol{\Xi}} \left[ \Phi(g(\{\mu_{c,i}\}_{c \in \mathcal{C}}, \{\sigma_{c,i}\}_{c \in \mathcal{C}}, \boldsymbol{\Sigma}_i^k, \xi_i, \boldsymbol{u}), \{\mu_{c,i}\}_{c \in \mathcal{C}}, \{\sigma_{c,i}\}_{c \in \mathcal{C}}) \right]. \tag{86}$$

From assumption 9 and equation 75, we have:

$$\mathbb{E}_{\boldsymbol{\Xi}} \left[ \frac{1}{d} \lambda r_d(\boldsymbol{G}_n) + s h_d(\boldsymbol{G}_n) \right] = q_n(s, \hat{\boldsymbol{u}}_n(s)). \tag{87}$$

Similar to the proof of Lemma 12, we obtain that $q_n(s, \boldsymbol{u})$ converges uniformly to $q(s, \boldsymbol{u})$ given by the corresponding expectation w.r.t the limiting empirical measure:

$$q(s, \boldsymbol{u}) = \mathbb{E}_{p(\boldsymbol{\sigma}, \boldsymbol{\mu})} \left[ \mathbb{E}_{\boldsymbol{\Xi}} \left[ \Phi(g(\{\mu_{c,i}\}_{c \in \mathcal{C}}, \{\sigma_{c,i}\}_{c \in \mathcal{C}}, \boldsymbol{\Sigma}_i^k, \xi_i, \boldsymbol{u}), \{\mu_{c,i}\}_{c \in \mathcal{C}}, \{\sigma_{c,i}\}_{c \in \mathcal{C}}) \right] \right]. \tag{88}$$

Due to the uniform convergence, we further have:

$$\lim_{n \to \infty} q_n(s, \hat{\boldsymbol{u}}_n(s)) = q(s, \hat{\boldsymbol{u}}(s)). \tag{89}$$

Using the above equation and Equation 83, we conclude that:

$$\mathbb{E}_{\boldsymbol{X}} \left[ \frac{1}{n} \sum_{i=1}^{n} \ell \left( \frac{\boldsymbol{W}^* \boldsymbol{x}_i}{\sqrt{d}} + \boldsymbol{b}^*, \boldsymbol{y}_i \right) + \frac{1}{d} \lambda r(\boldsymbol{W}^*) + s h(\boldsymbol{W}^*) \right] \to \sum_{k=1}^{K} \rho_k \mathbb{E}_{\boldsymbol{\xi}}[\ell(\boldsymbol{e}_k, \boldsymbol{h}_k(\hat{\boldsymbol{u}}(s)))] + q(s, \hat{\boldsymbol{u}}(s)), \tag{90}$$

where $\boldsymbol{h}_k$ and $q$ are differentiable functions of $\hat{\boldsymbol{u}}(s)$.

Since the RHS is a differentiable function in $s$, Assumption 5 is satisfied for the perturbation $h(\boldsymbol{W})$. Due to the coercivity of $\ell(\boldsymbol{y}, \bullet \boldsymbol{X}) + r(\bullet)$, there exists a sequence of fixed compact subsets containing the minimizers $\boldsymbol{W}^*$ with high probability as $n \to \infty$ (see Lemma 8 in [18]). Furthermore, since the input distribution is given by a mixture of gaussians with bounded means, Assumption 8 is satisfied for any such sequence of constraint sets. Therefore, the validity of assumption 5 through Equation 5 allows the applicability of Theorem 4 for the statistic $h_d(\boldsymbol{W})$. Through standard approximation technniques or the Stone–Weierstrass theorem, the restriction of differentiability and bounded Hessian of $\Phi$ can be removed. This completes the proof of Theorem 5 for general bounded Lipschitz $\Phi$.

## B  Assumptions on the target function

In this section, we discuss possible generalizations of the assumptions on the target function. In (2), we assume a target function depending on a small number of linear projections in the input space, along with the class labels. However, when the inputs are generated through feature maps $\boldsymbol{x} = \psi(\boldsymbol{z})$, one may instead consider target functions depending directly on the latent vectors $\boldsymbol{z}$. This was the setup considered in [16] for random feature maps. For mixture models considered in our work, one may assume:

$$y_i(\boldsymbol{X}) = \eta(\boldsymbol{\Theta}_\star^\top \boldsymbol{z}_i, \varepsilon_i, c_i). \tag{91}$$

We conjecture that our results can be generalized to the above setup through the use of the following stronger assumption:

**Assumption A6.** *For any Lipschitz function $\varphi : \mathbb{R} \to \mathbb{R}$,*

$$\lim_{n,p \to \infty} \sup_{\boldsymbol{\Theta}_1 \in \mathcal{S}_p^{\boldsymbol{x}}, \boldsymbol{\Theta}_2 \in \mathcal{S}_d^{\boldsymbol{z}}} \left| \mathbb{E} \left[ \varphi(\boldsymbol{\Theta}_1^\top \boldsymbol{x}, \boldsymbol{\Theta}_2^\top \boldsymbol{z}) \,\middle|\, c_{\boldsymbol{x}} = c \right] - \mathbb{E} \left[ \varphi(\boldsymbol{\Theta}_1^\top \boldsymbol{g}, \boldsymbol{\Theta}_2^\top \boldsymbol{z}) \,\middle|\, c_{\boldsymbol{g}} = c \right] \right| = 0, \quad \forall c \in \mathcal{C}. \tag{92}$$

*Here $S_p^{\boldsymbol{x}}$ is the constraint set for the training parameters $\boldsymbol{\Theta}_1$ while $\mathcal{S}_d^{\boldsymbol{z}}$ denotes a suitable constraint set on $\mathbb{R}^d$ where $d$ denotes the dimension of the latent vectors. Under the above assumption*

Such an assumption has been discussed in [13] under the term "Hidden Manifold Model", and was proven in [16] for random feature maps acting on Gaussian noise.

## C One-dimensional gaussian approximation

Although Theorem 7 is a powerful result, it still relies on very strong assumptions. In particular, given a distribution $\nu$ for the inputs $\boldsymbol{x}_i$, characterizing the set of vectors $\boldsymbol{\theta}$ such that Assumption 4 holds is in general a difficult task.

**Rigorous results** When the entries of $\boldsymbol{x}$ are i.i.d subgaussian, a classical application of the Lindeberg method [50] shows that Assumptions 2 and 4 are satisfied with

$$\mathcal{S}_p = \{\boldsymbol{\theta} \in \mathbb{R}^p \mid \|\boldsymbol{\theta}\|_\infty = o_p(1)\}.$$

More recently, this result (often used under the name "Gaussian Equivalence Theorem") was extended to general feature models with approximate orthogonality constraints [16, 14], for the same choice of $\mathcal{S}_p$. [17] also provides a central limit theorem result for the Neural Tangent Kernel of [51], for a more convoluted parameter set $\mathcal{S}_p$. While these papers provide a strong basis for the one-dimensional CLT, those rigorous results only concern (so far) a very restricted set of distributions.

**Concentration of the norm** Another, more informal line of work originating from [23], argues that most distributions found in the real world satisfy some form of the central limit theorem. The starting point of this analysis is the following theorem, adapted from [52]:

**Theorem 14** (Corollary 2.5 from [52]). *Let $\boldsymbol{x} \in \mathbb{R}^p$ be a random variable, with $\mathbb{E} * \boldsymbol{x}\boldsymbol{x}^\top = \boldsymbol{I}_p$, and $\eta_p$ the smallest positive number such that*

$$\mathbb{P}\left(\left|\frac{\|\boldsymbol{x}\|_2}{\sqrt{p}} - 1\right| \geq \eta_p\right) \leq \eta_p. \tag{93}$$

*Then for any $\delta > 0$, there exists a subset $\mathcal{S}_p$ of the $p$-sphere $\mathbb{S}^{p-1}$ of measure at least $4p^{3/8}e^{-cp\delta^4}$, such that*

$$\sup_{\boldsymbol{\theta} \in \mathcal{S}_p} \sup_{t \in \mathbb{R}} \left|\mathbb{P}(\boldsymbol{\theta}^\top \boldsymbol{x} \geq t) - \Phi(t)\right| \leq \delta + 4\eta_p,$$

*where $\Phi$ is the characteristic function of a standard Gaussian, and $c$ is a universal constant.*

If both $\delta$ and $\eta_p$ are $o(1)$, Theorem 14 implies that Assumption 4 is satisfied for any compact subset $\mathcal{S}'_p \subseteq \mathcal{S}_p$. This suggests that the norm concentration property of (93) is a convenient proxy for one-dimensional CLTs. However, the proof of this theorem uses isoperimetric inequalities, and is thus non-constructive; as a result, characterizing precisely the set $\mathcal{S}_p$ remains an open and challenging mathematical problem.

**Concentrated vectors** In [23], the authors consider the concept of *concentrated* random variables, as defined in [53]:

**Definition 15.** *Let $\boldsymbol{x} \in \mathbb{R}^p$ be a random vector. $\boldsymbol{x}$ is called (exponentially) concentrated if there exists two constants $C, c$ such that for any 1-Lipschitz function $f : \mathbb{R}^p \to \mathbb{R}$, we have*

$$\mathbb{P}(|f(\boldsymbol{x}) - \mathbb{E}f(\boldsymbol{x})| \geq t) \leq Ce^{-ct^2}.$$

Since the norm function is 1-Lipschitz, it can be shown that any concentrated isotropic vector $\boldsymbol{x}$ satisfies (93), with

$$\eta_p \propto \left(\frac{\log(p)}{p}\right)^{1/2}$$

The converse is obviously not true; an exponential random vector still has $\eta_p \to 0$, but is not concentrated. However, even if it is stronger that (93), the concept of concentrated vectors has two important properties:

(i) a standard Gaussian vector $\boldsymbol{x} \sim \mathcal{N}(\boldsymbol{0}, \boldsymbol{I}_p)$ satisfies Definition 15 with constants $C, c$ independent from $p$,

(ii) if $\boldsymbol{x} \in \mathbb{R}^p$ is a concentrated vector with constants $C, c$ and $\Psi : \mathbb{R}^p \to \mathbb{R}^q$ is an $L$-Lipschitz function, then $\Psi(\boldsymbol{x})$ is also a concentrated vector, with constants only depending on $c, C$ and $L$.

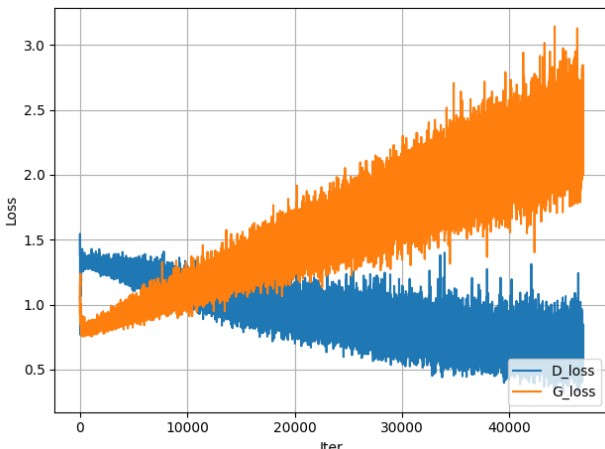

Figure 3: Evolution of the binary cross entropy training loss for the generator (orange) and discriminator (blue) during training.

**Towards real-world datasets**    The real-world data considered in machine learning is often composed of very high-dimensional inputs, corresponding to $p \gg 1$ in our setting. However, it is generally accepted that this data actually lies on a low-dimensional manifold of dimension $d_0$: this is the idea behind many dimensionality reduction techniques, from PCA [54] to autoencoders [55]. Another, more recent line of work (see e.g. [56]) studies the estimation of the latent dimension $d_0$; results for the MNIST dataset ($p = 784$) yield $d_0 \approx 15$, while CIFAR-10 ($p = 3072$) has estimated intrinsic dimension $d_0 \approx 35$ [57].

Following this heuristic, the most widely used method to model realistic data is to learn a map $f : \mathbb{R}^{d_0} \to \mathbb{R}^p$, usually through a deep neural network, and then generate the $x_i$ according to

$$\boldsymbol{x} = f(\mathbf{z}) \quad \text{with} \quad \mathbf{z} \sim \mathcal{N}(\mathbf{0}, \boldsymbol{I}_{d_0}) \tag{94}$$

Examples of functions $f$ include GANs [58], variational auto-encoders [59], or normalizing flows [60]. This ansatz has been studied theoretically, and the results compared with real-world datasets, in [13, 18]; the results indicate significant agreement between generated inputs and actual data.

Finally, we argue that for a large class of generative networks, the learned function $f$ is actually Lipschitz, with a bounded constant. This is even often a design choice; indeed, theoretical results such as [61] imply that a smaller Lipschitz constant improve the generalization capabilities of a network, or its numerical stability [62]. As a result, regularizations aimed at controlling the Lipschitz properties of a network are a common occurrence; see e.g. [63] for the spectral regularization of GANs. This indicates that concentrated vectors are indeed a good approximation for real-world data.

## D    Details on the numerical simulations

In this appendix we expand on how Fig. 2 was generated. This closely follows the pipeline illustrated in Fig. 1.

**Step 1: Training of the cGAN —**    The first step consists on training a cGAN on the real data set. For Fig. 2, we have used a PyTorch [64] implementation of the architecture in [65] publicly available at the pytorch-generative-model-collections repository. The cGAN was trained on the fashion-MNIST dataset [45] following the default procedure in the repository: i.e. training for 50 epochs and batch size 64 using Adam with learning rate 0.0002 and $(\beta_1, \beta_2) = (0.5, 0.999)$ on the binary cross entropy (BCE) loss (equal for both generator and discriminator). The evolution of the training loss during training is given in Fig. 3, and samples from the generator during different epochs are shown in Fig. 4.

**Step 2: Evaluating the class means and covariances —**    With the trained cGAN in hands, we can generate as many fresh samples as needed for our experiments. Moreover, a feature map can

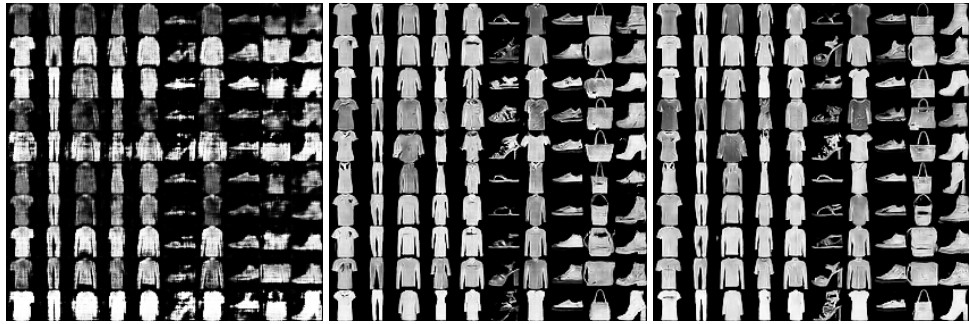

Figure 4: Fake fashion-MNIST Samples from the cGAN generator at the end of epoch 1 (left), 25 (middle) and 50 (right).

be easily added on the top of the cGAN architecture, as illustrated in Fig. 1. For Fig. 2, we have added a random feature map [46] $x \mapsto \tanh(Fx)$ with projection matrix $F \in \mathbb{R}^{1176 \times 784}$ with entries $F_{ij} \sim \mathcal{N}(0, 1/d)$. In order to compare the performance of a model trained on cGAN+RF samples vs. the equivalent Gaussian mixture model, we need to compute the class-wise means and covariances $(\mu_c, \Sigma_c)$. For Fig. 2, this was done with a standard Monte Carlo scheme over $10^6$ samples.

**Step 3: Learning curves —** The last step consists of computing the curves for the test error of logistic and ridge regression trained on the cGAN+RF features. Each point in Fig. 2 corresponds to a fixed sample complexity $\alpha = n/p$. For each $\alpha$, we generate fresh $n = \alpha \times p$ training features either from the cGAN+RF model (blue points) or from the equivalent Gaussian mixture model (red points). For the binary classification task, we split the samples over even vs. odds class labels. The SciPy [66] implementation of both ridge and logistic regression were used to train a classifier on the training data, from which both training error and test error were computed, using another batch of fresh samples for the latter. Finally, to reduce finite-size effects this procedure was repeated over 10 different seeds, with the average and standard deviation reported in Fig. 2.