# OpenReview forum: "Universality laws for Gaussian mixtures in generalized linear models"
_NeurIPS.cc/2023/Conference — NeurIPS 2023 poster_

### Official Review · Reviewer_tZtP · 2023-06-27

**Soundness:** 4 excellent
**Presentation:** 3 good
**Contribution:** 4 excellent
**Rating:** 7
**Confidence:** 3

**Summary:**

In classification problems with a finite number of classes under a supervised setting, if the feature vectors are generated from a mixture of distributions conditioned on the class labels and the learning is performed using generalized linear models, a phenomenon called ``Gaussian mixture equivalence (GME)'' simplifying the problem is known to occur. GME is the phenomenon in the high-dimensional limit that when considering a model in which the original conditional distribution of the feature vector is replaced by a Gaussian distribution having the same first- and second-order moments as the original ones, the quantities of interest such as generalization error coincide between the original model with the non-trivial class-conditional distribution and the reduced model with the Gaussian distribution. This paper rigorously proves this GME actually occurs under some (partly mild, partly strong) conditions by extending the method of reference [17] to the case involving multiple sets of parameters, which may be determined from the minimization of different objectives or by sampling from different Gibbs distributions.


**Strengths:**

Originality & Significance: Although GME has been assumed in previous studies and meaningful results have been shown under the assumption, the results of this paper clarifies the sufficient conditions under which GME can actually occur by the rigorous proof. This provides a mathematical foundation for attempts to understand complicated machine learning/statistical models' behavior through the analysis of simple Gaussian mixture models. The originality and the significance are thus high.


Clarity & Quality: The main text mainly provides the statements about the theorems, assumptions, and their consequences/applicabilities, and most of the proof details, even including the intuitive arguments, are deferred to the appendix. The statements are also not easy to understand, because many things are explained in the limited pages and the relationships among the theorems, assumptions, and consequences/applicabilities are not evident. The distinction between what has been proven in previous studies and what has been reiterated here is also not clear. Hence the clarity is not high. Nevertheless, the argument can be understood if one reads the manuscript carefully, and considering the importance of what is proven, also with the nice numerical experiment completely supporting the proposition, it is considered that the quality is high.


Here I enumerate some typos found and uneasy expressions for better clarity:

Line 162: The sentence is not easy to understand.

Below Line 200: The quantity $\ell'$ is not explained.

Line 225: be be -> be

Line 246: Theorem 2 -> Corollary 2

Line 262: a result convex -> a result of convex

Line 305,325: gaussian -> Gaussian


**Weaknesses:**

- Proof technicalities seem to much rely on [17], and there does not appear to have been much technical progress when extending it to the current case.

- Assumption 4 is the key assumption in the proof but when it does hold is not clear. Maybe this is the difficult point of the problem, and hence some more explanations in words are welcome.


**Questions:**

- Is it possible to extend the proof to the case where $\bm{x}_i$ has correlations among $i \in [n]$? An interesting example is the orthogonal ensemble under $p>n$. Some discussions are welcome.


**Limitations:**

The authors address their research limitations well. I think there is no concern about the potential societal impact.

---

> ### Author Rebuttal · Authors · 2023-08-09
>
> We thank the reviewer for their comments and suggestions which will help improving this work. Below, we address specific comments and questions:
>
> > Proof technicalities seem to much rely on [17], and there does not appear to have been much technical progress when extending it to the current case.
>
> We refer to the general response regarding the technical and conceptual novelty of our work.
>
> > Assumption 4 is the key assumption in the proof but when it does hold is not clear. Maybe this is the difficult point of the problem, and hence some more explanations in words are welcome.
>
> As discussed in the reply to *Reviewer JkJQ* and explained in Appendix A.2, the key idea is to split the proof of universality in two steps. First, one shows that asymptotic universality of an observable $h$ can be reduced to the proof of a one-dimensional CLT. Second, one proves this CLT holds for the particular class of features of interest.
>
> This proof scheme streamlines universality proofs. Our work provides a general proof of the first step (Theorem 4) conditioned on the second (Assumption 4), later showing that Assumption 4 holds for some natural feature maps of interest, e.g. random features applied to a Gaussian mixture (Theorem 6). However, Theorem 4 holds independently, cutting in half the work of proving universality for other feature maps.
>
> We will clarify the importance of this assumption in the revised version.
>
> > Is it possible to extend the proof to the case where $x_{i}$ has correlations among $i\in[n]$? An interesting example is the orthogonal ensemble under $p>n$. Some discussions are welcome.
>
> This is a very interesting question ! On one hand, the iid property of the $x_i$ is crucial to ensure that eq. (34) is an iid sum, and hence that we can bound each term separately; extending it to correlated data requires a major innovation in the proof technique.
>
> On the other hand, assume that $(x_i)\_{i\in [n]}$ are random orthogonal vectors. For a given $\theta$, the random vector $(\theta^\top x_i)\_{i\in [n]}$ is a random projection of a spherical vector. Since this distribution converges weakly to a Gaussian, and the empirical risk $\hat{\mathcal{R}}\_n(\theta; x, y(x))$ is a Lipschitz function in $(\theta^\top x_i, \theta^{\star \top} x_i)\_{i\in [n]}$, we might expect that:
> $$\hat{\mathcal{R}}\_n(\theta; X, y(X)) \approx \hat{\mathcal R}\_n(\theta; G, y(G)),$$
>
> where $G$ is a matrix of i.i.d Gaussians, which is the crucial first step to prove universality. This property may also hold for other choices of distribution for $X$, as long as the gaussian equivalent $G$ is chosen accordingly. Hence, even though the proof does not seem to translate that well to the weakly dependent setting, we expect that a suitably defined universality property can also hold.

---

> > ### Comment · Reviewer_tZtP · 2023-08-15
> >
> > Thank you for providing a nice clarification on the difference between the presented result and the earlier known results. I think this point should be more clearly stated in the revised version. I also thank the answer to my question about the correlated case. I think this point is better to be in the revised paper. I support the acceptance of the paper by keeping the current score.

---

### Official Review · Reviewer_Wbxr · 2023-07-04

**Soundness:** 3 good
**Presentation:** 4 excellent
**Contribution:** 3 good
**Rating:** 6
**Confidence:** 4

**Summary:**

In this paper, the authors investigated the joint statistical behaviors of a set of genealized linear estimators obtained from either empirical risk minimization or sampling, for data and labels drawn from a mixture model, in the high-dimensional regime where where the data dimension $p$ and the sample size $n$ both go to infinity.

In particular, they establish sufficient conditions under which the asymptotic joint statistics of this family depend only on the means and covariances of the mixture model.

The obtained results extend a few previous efforts: for instance, provide alternative proof to the fact that Gaussian mixture observed through a random feature map is akin to a Gaussian mixture [11, 12, 21, 22, 16]; and that data generated by conditional Generative Adversarial Networks (cGAN) are like Gaussian mixture [23], etc.

Some theoretical results are presented in a not-so-precise manner and some clarifications are needed. The paper is in general well written otherwise.


**Strengths:**

The topic focuses on the emerging topic of universality of ML methods in the high-dimensional regime.
The obtained theoretical results are strong and improve prior arts in many ways.
I did not check the detailed proofs, but the proof sketch look compelling.

**Weaknesses:**

There are a lot of assumptions in the paper, and it somewhat makes the paper less easy to grasp and compare to existing works, some more explanations are needed.
Some clarifications are needed to better understand the statements in the paper, as well as the major contribution of this paper.


**Questions:**

I have the following minor comments or questions:

1. Assumption 2 indeed describes the type of (mixture) data distribution under study, some more explanation and/or comparison would be helpful here.
2. Corollary 2 seems to be a corollary of Theorem 1, and if this is the case, it should be announced to hold under "the same setting of Theorem 1" or under "Assumption 1-4".
2. Theorem 1 and Corollary 2 hold as long as one takes the limit of $n,p \to \infty$? Should some assumption be put on their ratio $p/n$?
3. last paragraph in page 6 "this would require a diverging number of Gaussian in the mixture": would the authors elaborate on this, with some reference or more details for example?
4. it would be helpful to comment on the necessity and/or importance of Assumption 7-9, otherwise they are a bit difficult to digest and compare.
5. Also, the dimension $n,p$ are NOT involved in Theorem 5?
6. as expectations w.r.t the joint empirical measure in 24 -> in (24) or Equation 24.
7. could the authors elaborate on the replica equations in Theorem 5: can they be numerically evaluated (with ease)?

**Limitations:**

This work is theoretical and I do not see any potential negative societal impact.

---

> ### Author Rebuttal · Authors · 2023-08-09
>
> We thank the reviewer for their comments and suggestions which will help improving this work. Below, we address specific comments and questions:
>
> > There are a lot of assumptions in the paper, and it somewhat makes the paper less easy to grasp and compare to existing works, some more explanations are needed. Some clarifications are needed to better understand the statements in the paper, as well as the major contribution of this paper.
>
> >1. Assumption 2 indeed describes the type of (mixture) data distribution under study, some more explanation and/or comparison would be helpful here.
>
> Assumption 2 guarantees that several quantities of interest throughout the proof stay bounded with high probability.
> We do not consider this assumption as inherently limiting, it should be possible to relax it through the use of a different proof technique. For instance, the bounded norm condition on the means $\mu_c$ can be removed for coercive losses or with strongly convex regularization, since without loss of generality the constraint set can be restricted to parameters such that $\theta^\top \mu_c$ remains bounded. We will add further details and explanations in the final version.
>
> >2. Corollary 2 seems to be a corollary of Theorem 1, and if this is the case, it should be announced to hold under "the same setting of Theorem 1" or under "Assumption 1-4".}}
>
> Thanks for pointing this out. We will update the statement of Corollary 2.
>
> > 3. Theorem 1 and Corollary 2 hold as long as one takes the limit of $n,p\to\infty$? Should some assumption be put on their ratio $n/p$?
>
> > 6. Also, the dimension $n,p$ are NOT involved in Theorem 5?
>
>
> All of our results hold for $n/p \rightarrow \alpha$ for some constant $\alpha$: this was mentioned in the appendix, but we will emphasize it in the main text. Regarding Theorem 5, this is a typo; $d$ should be replaced by $p$ in the statements of section 2.3.
>
>
> > 4. last paragraph in page 6 "this would require a diverging number of Gaussian in the mixture": would the authors elaborate on this, with some reference or more details for example?
>
> We do not have a proof that a specific family distribution actually requires a diverging number of Gaussians to approximate; however, the Stein-Weierstrass theorem does not entail dimension-independent bounds on the number of Gaussians required, hence those types of results are not enough to ensure that Assumption 4 holds.
>
> > 5. it would be helpful to comment on the necessity and/or importance of Assumption 7-9, otherwise they are a bit difficult to digest and compare.
>
> We can classify Assumptions 7-9 into three categories:
> - The boundedness of the $\Sigma_c$ and $\mu$ simply guarantee that we are in a case covered both by [12] and by the assumptions of Theorem 4.
> - The diagonal property of the $\Sigma_c$, as well as the joint convergence, ensure that we can view the minimization problem (22) as a finite-dimensional approximation of an infinite-dimension problem. This guarantees that there is a well-defined object that $W^\star, b^\star$ will converge towards.
> - Finally, the separability assumption on $r$ (which is satisfied by most regularizers in practice, e.g. any $\lVert \cdot \rVert_p^p$) responds to the fact that we aim for a result on the empirical coordinate distribution of $W^\star, b^\star$; hence we need to be able to express $r$ only as a function of this empirical distribution.
>
>
> > 8. could the authors elaborate on the replica equations in Theorem 5: can they be numerically evaluated (with ease)?
>
> The replica equations are borrowed from the work of [12](Loureiro et al. (2021)) which derived them from the state evolution of an Approximate Message Passing scheme for generalized linear models trained on Gaussian mixtures. Despite looking cumbersome, these are scalar equations which are easy to evaluate numerically.

---

> > ### Comment · Reviewer_Wbxr · 2023-08-14
> >
> > I thank the authors for their clarifications. And I maintain my positive rating on the paper.

---

### Official Review · Reviewer_KxLS · 2023-07-06

**Soundness:** 2 fair
**Presentation:** 1 poor
**Contribution:** 2 fair
**Rating:** 3
**Confidence:** 4

**Summary:**

The submission studies the universality phenomena for general mixture model. The authors show the universality of empirical risk minimization and sampling and several extensions.

**Strengths:**

The authors give good introduction and literature review.

**Weaknesses:**

1. The main contribution lies in the technical part. However, it lacks novelty given the results in Hu and Lu 2022, Montanari and Saeed 2022. The assumptions 1-4 and Section 2.1 are almost identical to the main results in Montanari and Saeed 2022 without mentioning or comparisons at the beginning. The proofs for Theorem 1 and Corollary 2 only recap the previous results in the appendix. Section 2.2 gives some simple extensions of those results in Montanari and Saeed 2022 to multiple risk case.

2. The paper has plenty of inconsistencies and lack of explanations in the course of writing. For example, line 109, the notation of \eta is not consistent to equation (2). The three applications in Section 1 are hard to follow given the model setting.

3. Some of the notations are not proper: For example, line 101, it would be better to write \hat{y}=F(\hat{\Theta}^T x) to denote estimated parameters. I suggest the authors modify the notations in the whole paper to fit their own topic and interests instead of using those in Montanari and Saeed 2022.

4. Also, it would be better to explain the notations immediately after introducing.

5. The statements of the main theorems and proofs in the appendix is lack of polish. It would be better to write remarks after the theorems to improve the readability.


**Questions:**

Could the authors explain in more details the technical differences between their submission and the previous analyses?

---

> ### Author Rebuttal · Authors · 2023-08-09
>
> We thank the reviewer for their comments and suggestions which will help improving this work. Below, we address specific comments and questions:
>
> > 1. The main contribution lies in the technical part. However, it lacks novelty given the results in Hu and Lu 2022, Montanari and Saeed 2022. The assumptions 1-4 and Section 2.1 are almost identical to the main results in Montanari and Saeed 2022 without mentioning or comparisons at the beginning. The proofs for Theorem 1 and Corollary 2 only recap the previous results in the appendix. Section 2.2 gives some simple extensions of those results in Montanari and Saeed 2022 to multiple risk case.
>
> > Could the authors explain in more details the technical differences between their submission and the previous analyses?
>
> We refer the reviewer to the general response regarding a summary of the technical and conceptual novelty of our work. To address their specific points:
>
> - Section 2.1, and our Assumptions 1-4, generalize Assumptions 1-5 and Theorem 1 of [17] (Montanari and Saeed (2022)) from a single Gaussian to the Gaussian mixture setting as in eqs. (1),(8). The main differences are outlined in lines 633-645 in Appendix A. The most important distinction is in point (ii), where we allow the labeling function to depend on the mixture label of each datapoint.
> Although our proof omits the technical details identical to the ones in [17](Montanari and Saeed (2022)), the proof does require considerable adaptations.
>    In particular, we introduce an additional dependency by coupling the mixture labels of the datapoints $x_i$ and their Gaussian counterparts, that we handle by controlling the terms corresponding to each mixture separately.
>    This is detailed in Section A.4 of the appendix (lines 627-632 and 662-664).
> - About section 2.2. What is considered "simple" is of course in the eye of the beholder. We respectfully disagree: constructing the reduction from multiple parameters, such as required in Bayesian inference and ensembling, to a single equivalent Boltzmann distribution on a lifted parameter space (Equation (49) in the Appendix) is technically non-trivial from our point of view. This leads to our theorem 4 which has a much broader applicability than the ones in previous literature. We refer to concrete such cases in the common answer. Furthermore, as we mentioned in the general response, our proof of Theorem 4 relies on a novel convexity-based argument absent in earlier works.
>
> > 2. The paper has plenty of inconsistencies and lack of explanations in the course of writing. For example, line 109, the notation of $\eta$ is not consistent to equation (2). The three applications in Section 1 are hard to follow given the model setting.
>
> > 3. Some of the notations are not proper: For example, line 101, it would be better to write $\hat{y}=F(\hat{\Theta}^T x)$ to denote estimated parameters. I suggest the authors modify the notations in the whole paper to fit their own topic and interests instead of using those in Montanari and Saeed 2022.
>
> We will modify some of the notation in our paper to improve readability and consistency.
> For clarification, the definition of $\hat{y}$ in line 101 is applicable for any $\Theta$ and is used to define the empirical risk objective in line 103. The estimated parameters $\hat{\Theta}$ are introduced only later in equation 4.
>
> > 4. Also, it would be better to explain the notations immediately after introducing.
>
> > 5. The statements of the main theorems and proofs in the appendix is lack of polish. It would be better to write remarks after the theorems to improve the readability.
>
> We thank the reviewer for the suggestions. We will add explanations for the notations and remarks after the theorems in the final version.

---

> > ### Comment · Reviewer_KxLS · 2023-08-14
> >
> > Thanks for the authors' response.
> >
> > I agree that the theorems in the submission require considerable adaptations. However, the technical novelty seems to be still limited.
> >
> > I totally understand the meanings of $\hat{y}$ and $\hat\Theta$ in the main text. What I recommended is that you would better adopt a system of consistent notations. Hat version normally refers to statistics or estimators involving samples, which is not proper to denote dummy quantities. Please adopt a coherent framework which uses widely accepted convention within the community instead of grafting onto someone else's work.
> >
> > Without the authors carefully revising the main text, I would still maintain my ratings.

---

> > > ### Author Response · Authors · 2023-08-14
> > >
> > > As anticipated in the rebuttal, we will carefully revise the notation and take into account all the valuable feedback. Unfortunately, the manuscript cannot be updated during the discussion period. In particular, we will use different notation for $y$ in line 101 (and the abstract) and reserve the $\hat{y}$ notation for estimators evaluated from the data

---

### Official Review · Reviewer_JkJQ · 2023-07-06

**Soundness:** 3 good
**Presentation:** 3 good
**Contribution:** 3 good
**Rating:** 7
**Confidence:** 3

**Summary:**

The paper proves that the performance and other quantities of interest output by a number of algorithms in the generalized linear models class are the same as if the data were replaced by Gaussians with the same means and covariance matrices. This universality result suggests that the performance of this entire class of algorithms (generalized linear models) depends only on the first- and second-order moments of the data. This theoretical study justifies the use of a Gaussian mixture model in the study of generalized linear algorithms.
The particularity of the study consists in proving this important phenomenon rigorously even on restrictive assumptions.

**Strengths:**

1-The question of the universality of Gaussian mixtures in the analysis of the performance of generalized linear algorithms is a subject that has long remained empirical and has known very few rigorous and clear answers. This paper by providing a rigorous answer, makes it possible to make an important step in the resolution of this question.

2- The paper is generally well written with very few spelling mistakes and consistency in notations. The motivation and structure of the paper seems to be quite clear.

3- Although not having checked the theoretical calculations, the approach seems to be rigorous and well conducted. The results are consistent.

**Weaknesses:**

1- Assumption 4 and Assumption 5 need to be better explained in terms of their intuitive meaning and how they are binding. Although there is some discussion that it is very restrictive and experiments tend to suggest that the results should be valid beyond that, it would be important to provide more intuitive explanations of what it means and why it is restrictive.

2- In the experiment shown in Figure 2, there is a systematic bias between the performance of the CGAN and the GMM. It would be interesting to discuss this aspect, providing explanations if necessary. It would have been useful to show the limits of universality in a general way (even in the appendix). Are there cases where this holds less well, and this would be due to what phenomenon, or are there algorithms that are more resistant to this universality in the class of linear algorithms?

3- Although I appreciate the authors' effort to provide experiments on real data, it would have been interesting to do so with different data structures and slightly more challenging data such as time series and language classification. In particular, we could ask the question of universality for highly structured data such as very sparse data (like speech encoding) or time series. It seems somewhat counter-intuitive that for these highly structured data, the mean and variance alone are sufficient to characterize performance. Recently, in random matrix theory, universality has been focused on a class of random vectors that come from the concentration of the measure and it is not obvious that highly sparse data are concentrated and therefore that universality holds for this type of data, even through the prism of a linear model.



**Questions:**

1- What is the intuitive meaning of Ass 4 and Ass 5? How restrictive they are?

2- Is the universality behave the same whatever the type of structure considered (highly sparse data, time series)? Do some experiments could be provided on some of these datasets? Are these some cases of failure of the universality noticed in experiments?

**Limitations:**

The main statement of the paper seems quite strong even though the mathematics seems rigorous. In this case, it would be really important to provide a wide set of experiments over several datasets (with different structure) and several linear algorithms. This would be more reassuring than checking the calculations line by line for the reader.

---

> ### Author Rebuttal · Authors · 2023-08-09
>
> We thank the reviewer for their comments and suggestions which will help improving this work. Below, we address specific comments and questions:
>
> > Assumption 4 and Assumption 5 need to be better explained in terms of their intuitive meaning and how they are binding. Although there is some discussion that it is very restrictive and experiments tend to suggest that the results should be valid beyond that, it would be important to provide more intuitive explanations of what it means and why it is restrictive.
>
> > What is the intuitive meaning of Ass 4 and Ass 5? How restrictive they are?
>
> Assumptions 4 and 5 conceptually play a very different role in the proof of Theorem 4.
>
> On one hand, Assumption 5 is a technical restriction, mainly stemming from the convexity based argument used to prove Theorem 4. We expect it to work in a fairly generic setting, and we show it also holds in the setting of Theorem 5.
>
> On the other hand, Assumption 4 plays a crucial role. As explained in Appendix A.2, the key idea is to split the proof of universality in two steps. First, one shows that asymptotic universality of an observable $h$ can be reduced to the proof of a one-dimensional CLT. Second, one proves this CLT holds for the particular class of features of interest.
>
> This proof scheme streamlines universality proofs. Our work provides a general proof of the first step (Theorem 4) conditioned on the second (Assumption 4), later showing that Assumption 4 holds for some natural feature maps of interest, e.g. random features applied to a Gaussian mixture (Theorem 6). However, Theorem 4 holds independently, cutting in half the work of proving universality for other feature maps.
>
> We will clarify the relative importance of each assumption in the revised version.
>
>
>
>
>
> > 2- In the experiment shown in Figure 2, there is a systematic bias between the performance of the CGAN and the GMM. It would be interesting to discuss this aspect, providing explanations if necessary. It would have been useful to show the limits of universality in a general way (even in the appendix). Are there cases where this holds less well, and this would be due to what phenomenon, or are there algorithms that are more resistant to this universality in the class of linear algorithms?
> }}
>
> A possible explanation for the systematic bias in Figure 2 is that performing classification using linear classifiers on GMM data with the same mean, covariance as the cGAN data is ``easier" since it only requires capturing the conditional first two moments. Indeed, we expect assumption 4 to only hold approximately for real data. Additionally, our results are asymptotic and can suffer from finite size effects which usually translate to a systematic bias.
>
> We agree with the reviewer that understanding the limitations of universality is an interesting and important problem. We will add further discussions about it in the final version.
>
>
> > 3- Although I appreciate the authors' effort to provide experiments on real data, it would have been interesting to do so with different data structures and slightly more challenging data such as time series and language classification. In particular, we could ask the question of universality for highly structured data such as very sparse data (like speech encoding) or time series. It seems somewhat counter-intuitive that for these highly structured data, the mean and variance alone are sufficient to characterize performance. Recently, in random matrix theory, universality has been focused on a class of random vectors that come from the concentration of the measure and it is not obvious that highly sparse data are concentrated and therefore that universality holds for this type of data, even through the prism of a linear model.
>
> > 2- Is the universality behave the same whatever the type of structure considered (highly sparse data, time series)? Do some experiments could be provided on some of these datasets? Are these some cases of failure of the universality noticed in experiments?
>
> > The main statement of the paper seems quite strong even though the mathematics seems rigorous. In this case, it would be really important to provide a wide set of experiments over several datasets (with different structure) and several linear algorithms. This would be more reassuring than checking the calculations line by line for the reader.
>
> Our universality results are as strong as the one-dimensional CLT. In particular, since one-dimensional projections of (very) sparse data are not Gaussian, we expect universality to break down in this case. For time series, given the high correlation between data points, the chosen directions of $\mathcal S_p$ should probably satisfy strong conditions to ensure that Assumptions 2 and 4 hold.
>
> On the other hand, positive examples are already abound in the literature. For instance, [11-13, 19-21] all compare the results of their exact Gaussian predictions in various settings with the performances on real-world datasets, and find a very good agreement between the two. Our work aims at providing a general framework to justify those previous results, hence we only included an illustrative example.
>
> Finally, for a explicit synthetic example breaking universality, we refer the reviewer to the systematic investigation by Goldt and Ingrosso (*Data-driven emergence of convolutional structure in neural networks*, PNAS, 2022).

---

> > ### Comment · Reviewer_JkJQ · 2023-08-15
> >
> > I would like to thank the authors for their time and effort in responding to our various concerns.
> >
> > I think the main limitation of the work remains the restrictive nature of the hypotheses. But this does not detract from my positive perception of the article given the significant contribution. Furthermore, I would encourage the authors to discuss the limitations and failures of universality in more detail, if only by pointing out useful references and illustrating these failures with arguments to explain them.
> >
> > I stand by my positive assessment of the article.

---

### Author Rebuttal · Authors · 2023-08-09

# General Response

We thank the reviewers for their valuable comments and suggestions. We will incorporate them into a revised version of our paper. In our understanding, the main concern across the reviews is the significance and technical novelty of our work with respect with the literature. Below, we address this concern in detail.

## Significance

Our results build on the long line of works on the proof of the Gaussian equivalence (GE) Principle, starting from the work [14] (Goldt et al. (2022)) who proved a 1d CLT theorem for random features of Gaussian data analogous to Assumption 4 in our work. The universality of the empirical risk minimizer for the random features model was then proven by [16](Hu and Lu (2022)) for Gaussian design. [17](Montanari and Saeed (2022)) provided an alternative proof for GE which relaxed the conditions of [16](Hu and Lu (2022)) from strongly convex to almost convex losses.

However, none of these results cover the different settings in which GE was empirically observed to hold. Covering them is precisely the goal of our work. Concretely:

- While [16](Hu and Lu (2022)) (see eq. (8)) and [17](Montanari and Saeed (2022)) (see Ass. 5) only consider the case where the equivalent model is a single-mode Gaussian, our Theorem 1 and Corollary 2 encompasses the much richer Gaussian Mixture model. This allows us to provide theoretical grounding to the works [21](Gerace et al. (2022)), [22](Pesce et al. (2023)), [12](Loureiro et al. (2021)).
- Existing proofs only consider the empirical minimization setting (eq. (1) in [16], eq. (8) in [17]), for a single estimator. Since our Theorem 4 extends to any ensemble of ERM objectives and (non-necessarily optimal) Bayes estimators, our results are applicable to a wide range of problems such as ensembling [20](Loureiro et al. (2022)) or uncertainty quantification [19](Clarté et al. (2022)), [24](Clarté et al. (2022)).
- Our one-dimensional CLT result extends the existing work from the input being a single Gaussian of zero mean (Theorem 2 of [16] and Corollary 2 of [17]) to the input being a Gaussians mixture with general non-zero mean (our Theorem 6). These kind of random features were for example used in [11](Refinetti et al (2021)) for Gaussian mixture classification.

Further, our results bridge the works of [38](Louart et al. (2018)); [23,39](Seddik et al. (2020, 2021)) for universality of conditional generative models to Gaussian equivalence of the empirical risk minimizer.

Hence, our results fill a significant gap in the rigorous understanding of GE which **is not covered by existing works**.

## Technical novelty

Concerning mathematical novelty, our proofs build on many existing techniques but also introduce several new ones that we anticipate to be of independent interest:

- **Theorem 4**: While the technique of adding a perturbation to the loss is well known, our proof leverages tools from convex analysis to cut short an otherwise lengthy argument from existing works. This approach is more general, enabling the proof for a general class of Gibbs measures involving multiple objectives as well as the empirical convergence of other observables. Previous proofs were restricted to the particular case of the generalization and training errors.
- **Theorem 5**: The starting point of our proof is the non-trivial observation that the nonlinear system of equations in [12](Loureiro et al. (2021)) can be described in a self-consistent manner using the joint-empirical measure of the means and covariances of the mixtures. Leveraging novel topological and analytical arguments, we relate weak convergence to the differentiability assumption (Assumption 5) of Theorem 4. This argument bridges exact asymptotic results for specific distributions to the universality of the corresponding estimators in a larger class of distributions, and we believe it can be of independent interest.

To summarize, our results significantly extends previous literature and provides rigorous guarantees to existing heuristic results.

This discussion will be reflected in the revised manuscript.

---

### Author Response · Authors · 2023-08-16
**Summary**

We thank the reviewers for their overall positive assessment of our work and for the constructive comments which help us improve our paper. We will be implementing all their suggestions in a revised version. In particular, we will:

- Make a throughout revision of the text, correcting all the typos flagged and revising the notations.
- Add a discussion on the assumptions and how each one fit within the proof scheme.
- Add additional discussion on the limitations and failures of universality.
- Incorporate the “*General answer*” discussion on the significance and technical novelty of our contribution with respect to previous literature to the main text.

Please let us know if you have any additional comment or suggestion.

---

### Decision · Program_Chairs · 2023-09-21

**Decision:**

Accept (poster)

**Comment:**

The paper provides a theoretical answer to the empirical question about the universality of Gaussian mixtures in the analysis of the performance of generalized linear algorithms. The work is well-regarded for its theoretical rigor and is considered to be a step forward in the field. However, there are several points where the reviewers felt the paper could be improved.

**Strengths:**
- Theoretical Rigor: All reviewers appreciated the rigorous theoretical framework presented in the paper.
- Originality and Significance: Reviewers noted the work's contribution in providing a mathematical foundation for understanding complex ML/statistical models through Gaussian mixture models.
- Empirical Consistency: Reviewers found the results to be consistent and the empirical section promising.

**Weaknesses:**
- Assumptions: Multiple reviewers pointed out that certain assumptions (especially Assumptions 4 and 5) need more intuitive explanations. The restrictive nature of these assumptions should be elaborated upon.
- Clarity and Readability: Several reviewers noted that the paper is hard to follow at times due to its technical depth and use of assumptions. Some inconsistencies in notation and lack of clarity in writing were also noted.
- Novelty: One reviewer pointed out that much of the technical content and assumptions seem to recap or rely heavily on previous work, notably those by Montanari and Saeed (2022).  This point should be explained more clearly in the final version.
- Experimental Scope: Reviewers suggested expanding the range of data types and structures studied, to more convincingly demonstrate the universality claimed in the paper.  Also, the presentation of "applications of interest" in the current manuscript should be improved for clarity in the final version.

The authors are strongly encouraged to address the mentioned concerns in making the final version.